

# Maximization of the precipitation from tropical cyclones over a target area through physically based storm transposition

Mathieu Mure-Ravaud [1], Alain Dib [1], M. Levent Kavvas [1], and Elena Yegorova [2]

[1]University of California Davis
[2]U.S. Nuclear Regulatory Commission
*Correspondence to:* Mathieu Mure-Ravaud (mmureravaud@ucdavis.edu)

**Abstract.** Certain methods of Probable Maximum Precipitation (PMP) estimation such as the generalized estimation method involve the transposition to the target area of intense storms that occurred in adjacent areas. This storm transposition step is based on the assumption that it is possible to delineate a "meteorologically homogeneous region" (World Meteorological Organization, 2009) surrounding the target area so that the precipitation field of a storm that occurred within this region may be transposed over the target area. Differences between conditions at the storm site and those at the project basin, such as differences in topography and in the proximity to the moisture source, are accounted for by "transposition adjustments". In this article, a new method for the transposition of tropical cyclones (TCs) is presented. This method is fully physically based as it uses a regional atmospheric model (RAM) to reconstruct the intense precipitation field from a TC, thus crucially conserving the mass, momentum and energy in the system. In this transposition method, the initial vortex in the simulation initial conditions is first shifted spatially. More precisely, the TC at the simulation start date is first separated from its background environment, then shifted, and finally recombined with the background environment. Then, the RAM is run as usual to simulate the TC and its precipitation field. The new transposition method was applied to four hurricanes which spawned torrential precipitation in the United States, namely Hurricanes Floyd (1999), Frances (2004), Ivan (2004), and Isaac (2012), in order to maximize the 72-h accumulated precipitation depth over the drainage basin of the city of Asheville, N.C. It was observed that the precipitation fields changed in both structure and intensity after transposition. Besides, the tracks of the hurricanes were generally very sensitive to changes in the initial conditions, which is expected for a storm system whose dynamics is strongly nonlinear. In particular, a small change in the location of the initial vortex may result in a very different track, allowing the TC to go over the target area.

# 1 Introduction

The design of a large structure, such as a dam or a nuclear plant located near a stream, requires estimating how large a flood can be at its specific location. Several methods have been proposed for such an estimation. In most cases flood frequency




analysis (FFA) is the preferred approach. FFA analysis uses historical measured data, sometimes combined with paleoflood data[1], in order to reconstruct the flood frequency curve at a specific location along a stream. The frequency of extreme floods is then estimated by extrapolating the flood frequency curve for a return period beyond the available data by the use of some statistical distribution. Another approach for the design of large structures is to determine the Probable Maximum Flood (PMF), which represents the potential maximum runoff resulting from the most severe combination of hydrological and meteorological conditions that are considered reasonably possible for a particular drainage basin (Shalaby, 1994). As extreme floods are usually triggered by extreme precipitation, it is legitimate to start the investigation of extreme floods by looking for the most extreme precipitation events that can occur over the basin containing the structure. As a result, hydrologists and meteorologists developed the concept of Probable Maximum Precipitation (PMP), which is the greatest depth of precipitation for a given duration meteorologically possible for a design watershed or a given storm area at a particular location at a particular time of year, with no allowance made for long-term climatic trends (World Meteorological Organization, 2009). The most widely used methods of PMP estimation are (World Meteorological Organization, 2009):

- The local method (local storm maximization or local model);

- The transposition method (storm transposition or transposition model);

- The combination method (temporal and spatial maximization of storm or storm combination or combination model);

- The inferential method (theoretical model or ratiocination model);

- The generalized method (generalized estimation);

- The statistical method (statistical estimation).

These methods have been used for several decades in order to provide PMP estimates in several countries including the United Stated (Hershfield, 1961, 1965; Corrigan, 1999), China (Zhan and Zhou, 1984), India (Rakhecha and Soman, 1994; Kulkarni et al., 2010), Thailand (Tingsanchali and Tanmanee, 2012), and Spain (Casas et al., 2008, 2011). The main advantage of these methods is that they are in general relatively simple to apply and do not require significant computational resources. However, there are several drawbacks to using these methods. Some of these methods, such as Hershfield's statistical method, strongly depend on observation data. According to Nobilis et al. (1991), Hershfield's method misestimates PMP values if the observation data include outliers. Furthermore, according to Koutsoyiannis (1999), there is no plausible reason to consider the estimates from Hershfield's statistical method as PMP values as these estimates show no evidence of the upper limit of precipitation. Other methods are more physically based but tend to make unjustified simplifications. For example, the transposition method assumes that it is possible to identify a "meteorologically homogeneous region" around the target area which sets the transposition limits, so that, if a given storm occurred in this region, its precipitation field may be transposed to the target area for the purpose of PMP estimation. The application of the storm transposition method can be flawed to the extent that

---

[1]Paleoflood hydrology is the reconstruction of the magnitude and frequency of recent, past, or ancient floods using geological evidence (Kochel and Baker, 1982). This includes erosional landforms, sediments, damage to vegetation and high-water marks.





the transposed precipitation field is usually significantly different from the original precipitation field in terms of its structure and intensity, as we show in this article for the case of TCs. Another example is the generalized estimation, which involves a moisture maximization step. The moisture maximization model is a linearized meteorological model that maximizes severe precipitation by the ratio of the maximum to the actual precipitable water. Precipitable water is calculated using persisting 12-h or 24-h dew points at the surface based on the assumption of a saturated pseudo-adiabatic atmosphere (World Meteorological Organization, 2009). The application of this moisture maximization method can be flawed to the extent that the relationship between precipitable water and dew point temperature at the surface is nonlinear (Abbs, 1999).

Recently, new methods for PMP estimation have been developed. Ohara et al. (2011) and Ishida et al. (2014, 2015) proposed a physically based approach for the estimation of the PMP, which they called "maximum precipitation" (MP) to distinguish it from the traditional PMP. In this approach, they used a regional atmospheric model (RAM) to reconstruct, through dynamical downscaling, the precipitation fields associated with intense atmospheric rivers[2] in California. Contrary to the traditional PMP approaches mentioned previously, their method has the advantage of conserving the mass, momentum and energy in the simulation domains, since the atmospheric model solves numerically the governing equations for the conservation of these quantities. Using a RAM also allows taking into account explicitly the effects of certain features that may generate extreme precipitation over a given area. For instance, RAMs explicitly account for the topography, which has been shown to play a major role in the generation of heavy rainfall in certain geographical regions (Wu et al., 2002; Ge et al., 2010; Lin et al., 2010).

The maximization of the precipitation from atmospheric rivers over a target basin (Ohara et al., 2011; Ishida et al., 2014, 2015) consists of 1) shifting the atmospheric state variables at the boundaries of the simulation outer domain, and 2) setting the relative humidity at the boundaries of the simulation outer domain to 100%. The first step (shifting) brings the storm over the target area, while the second step (moisture maximization) further maximizes the precipitation over this target area. Ishida et al. (2014, 2015) successfully applied this method to three watersheds in Northern California, subject to intense precipitation from atmospheric rivers. This physically based precipitation maximization method through the shifting of the boundary conditions is well suited for the maximization of precipitation from atmospheric rivers because the simulation of atmospheric rivers is essentially a boundary value problem: the severe conditions responsible for intense precipitation such as large moisture transport penetrate the simulation outer domain through the boundaries. On the other hand, several studies (e.g. Zou and Xiao, 2000; Wu, 2001) have emphasized the importance of using a realistic initial vortex for the numerical simulation of a tropical cyclone (TC). To this extent, the simulation of a TC is more of an initial value problem than it is a boundary value problem. As such, this article presents a new method for the storm transposition of TCs.

To the authors' knowledge, this is the first study investigating a fully physically based method to maximize the precipitation from a TC over a given target area. Lee et al. (2017) investigated the effects of increasing the sea temperature and maximizing the moisture at the domain's boundaries on the precipitation caused by Hurricane Rusa (2002) in Korea. They showed that the storm simulated within the aforementioned framework (i.e. increased sea temperature and increased moisture at the boundaries) produced significantly more rainfall over Korea than the historical storm. However, an increase in the precipitation depth over the target area due to an increase in temperature and boundary moisture may occur only if the original TC already spawned

---

[2]An atmospheric river is a narrow corridor of concentrated moisture in the atmosphere.




significant precipitation over this target area, as it is the case for Hurricane Rusa over Korea. In the case where the original TC did not affect the target area, increasing sea temperature and boundary moisture is unlikely to move the storm to the target area. As a result, in the general case, modifying sea temperature and moisture at the model's boundaries cannot answer the question: what would have happened if a given TC passed over a specified target area?

Therefore, in this article, precipitation from a TC over a target watershed is maximized by shifting the initial vortex of the TC. Emphasis is put on precipitation because of its importance in the design of large structures such as dams and nuclear plants. However, the transposition method proposed in this article can also be used to investigate what would have been the wind field or any other atmospheric field if the TC happened to pass over the target area. We show that, due to the nonlinearity in the dynamics of a TC, a very small shift of the initial vortex can result in a significant change in the track of the storm, sometimes

by several hundreds of kilometers, allowing the intense precipitation field from an originally distant TC to move over the target area.

Section 2 provides the technical details regarding the transposition (i.e. shifting) of the initial vortex. Section 3 applies the transposition method to Hurricane Ivan (2004). Section 4 presents the results for the transposition of three other TCs, namely Hurricane Floyd (1999), Hurricane Frances (2004), and Hurricane Isaac (2012). Section 5 proposes a procedure for the physi-

cally based estimation of the PMP for a target area for which intense precipitation is caused by TCs, as it is usually the case in the Eastern United States for a sufficiently large area[3]. Finally, Section 6 offers conclusions and perspectives.

## 2   Description of the TC transposition method

Initial and boundary conditions used for dynamical downscaling with a RAM are usually obtained from the output of coarse-resolution reanalysis atmospheric data or of a general circulation model (GCM). This section presents a method to shift

the location of a TC in the initial conditions. The objective of this transposition is to modify the track of the storm so that its precipitation field moves over a specified target area.

The transposition of the TC in the initial conditions is performed by executing the following procedure:

1. Identify the location $(x_c, y_c)$ of the center of low pressure;

2. Identify the radius $R$ of the cyclone;

3. Remove the TC from the background atmospheric fields (geopotential height, wind velocity, relative humidity, temperature, surface pressure, etc.) by cutting off the inside of the circle of center $(x_c, y_c)$ and of radius $R$ from the original atmospheric fields;

4. Interpolate the background fields to the inside of the circle;

---

[3]According to Zurndorfer et al. (1986), the type of storm which will produce the rains of PMP magnitude over small basins ($< 100$ mi$^2$) in and near the Tennessee River Watershed is of the thunderstorm variety, whereas for larger basins ($> 100$ mi$^2$) the primary rain producing storms are more likely to be TCs or decadent TCs potentially interacting and combining with other systems.





5. Compute the perturbation fields by subtracting the background fields obtained in step 4 from the original fields. The perturbation fields are zero everywhere except inside the circle;

6. Shift the perturbation fields;

7. Add the shifted perturbation fields to the corresponding background fields obtained in step 4.

This procedure is illustrated in Fig. 4 for the surface zonal wind velocity (i.e. for the $x$-component of the surface wind field) in Hurricane Ivan (2004). In practice, the storm will not be perfectly axisymmetric so that the radius $R$ in step 2 may be defined as the radius of the circle that contains the region of influence of the TC. The size of this region of influence can vary from one atmospheric field to another (e.g. wind velocity field vs. temperature field) as well as with height (e.g. surface wind field vs. wind field on the 500 mbar surface).

The transposition method assumes that it is possible to separate the contribution of the TC from its background environment. Ideally, the shifting exercise should be performed while the TC is still over the ocean, far from land and especially from its location of landfall. Furthermore, the shifting exercise should be performed before the TC starts its extratropical transition, in which case the system starts developing characteristic features such as high asymmetry, loss of warm core, fronts, tilt away from vertical, expansion of the wind field, and strong interaction with the midlatitude westerlies and possibly with extratropical

systems (Chan and Kepert, 2010). After a TC starts its extratropical transition, it may be difficult to assess what part of the field is due to the TC and what part of the field is due to other systems.

    On the other hand, if the shifting exercise is performed too early during the life cycle of the TC, the initial vortex may be too weak, which will impact the quality of the simulations and the intensity of the storm in its mature stage. As a consequence, one needs to be particularly careful in choosing the simulation start date so that the best compromise is found between the

20 aforementioned restrictions regarding the initial TC.

    The interpolation method used in step 4 is now described. One can use other interpolation methods if desired, as long as they produce realistic background fields. Let us consider a given atmospheric field (e.g. the surface pressure field). We denote the original field by $F_1$ and the field after transposition by $F_2$. The interpolation is performed by executing the following procedure:

1. Consider a square of side $2a$ with $a > R$ (Fig. 1);

2. For every point $(x_i, y_i)$ lying within the circle of center $(x_c, y_c)$ and of radius $R$, that is to say the points for which $d_{ic} \equiv \sqrt{(x_i - x_c)^2 + (y_i - y_c)^2} \leq R$, go to step 3;

3. Assign to every point $(x_o, y_o)$ lying within the square but outside the circle a weight inversely proportional to the distance between $(x_o, y_o)$ and $(x_i, y_i)$ . This weight can be computed as $w_o = \mathcal{N} \left[ (x_o - x_i)^2 + (y_o - y_i)^2 \right]^{-n/2}$ where $\mathcal{N}$ is a normalizing factor ensuring that the sum of the $w_o$ is equal to 1 and $n$ is an arbitrary positive constant;

4. Define $F_{tmp}(x_i, y_i) \equiv \sum_o w_o F_1(x_o, y_o)$ where the summation is over all points $(x_o, y_o)$ identified in step 3, that is to say the points lying within the square but outside the circle;

5. Compute $F_2(x_i, y_i) = \alpha F_{tmp}(x_i, y_i) + (1 - \alpha) F_1(x_i, y_i)$.




The last step is a smoothing step ensuring that the interpolated field inside the circle matches smoothly with the original field outside the circle. In this study, we used $a = 1.1 \times R$, $n = 6$ and chose $\alpha$ such that $\alpha = 1$ if $d_{ic} \leq 0.75\,R$ and $\alpha = \exp\left(-\frac{d_{ic}-0.75R}{R}\right)$ if $d_{ic} > 0.75\,R$.

## 3  Transposition of Hurricane Ivan (2004)

According to Stewart (2004), Ivan was a classical, long-lived Cape Verde hurricane. It reached Category 5 strength three times on the Saffir-Simpson Hurricane Scale, and was the strongest hurricane on record that far south east of the Lesser Antilles. It caused considerable damage and loss of life as it passed through the Caribbean Sea. Ivan made landfall as a Category 3 hurricane just west of Gulf Shores, Alabama on September 16th, 2004. It spawned heavy precipitation ranging from 3-7 in depth along a large swath from Alabama and the Florida panhandle northeastward across the eastern Tennessee Valley and into
the New England area (Fig. 8c).

In this study, Hurricane Ivan was simulated with the Weather Research and Forecasting (WRF) model (Skamarock and Klemp, 2008) (Version 3.7). No observation was used for nudging or data assimilation (since the location of the storm is modified in the initial conditions), so that the model was only subject to the influence of the initial and boundary conditions. The initial and boundary conditions were obtained from the Climate Forecast System Reanalysis (CFSR; Saha et al., 2010).
CFSR is produced by the U.S. National Weather Service National Centers for Environmental Prediction (NWS/NCEP) at $0.5°$ $\times$ $0.5°$ spatial resolution and 6-h temporal resolution. Three nested domains were used for the simulations (Fig. 2). The spatial resolutions of the outer (i.e. parent) domain, intermediate domain, and inner domain are 45 km, 15 km, and 5 km, respectively. The outer domain is composed of $160 \times 120$ nodes (zonal direction $\times$ meridional direction) while the intermediate domain is composed of $154 \times 151$ nodes and the inner domain is composed of $256 \times 238$ nodes.

Two-way nesting was used for the simulations, meaning that the different domains (outer, intermediate and inner) are run simultaneously and communicate with each other. The top of the model in the vertical extent was taken at 50 mbar, with a total of 38 vertical layers. A time step of 3 min was used. A simple 1-dimensional ocean mixed layer model was used following that of Pollard et al. (1972). The parameterization schemes used for the simulation of Hurricane Ivan are given in Table 1. This combination of the parameterization schemes comes from the calibration of the WRF model for the reconstruction of Hurricane
Ivan, which is discussed in the appendix. Cumulus parameterization was used only in the outer and intermediate domains. The simulation start date is 09/06/2004 00:00 UTC. At that time, Hurricane Ivan was located off the coasts of French Guiana and Suriname (see Figs. 4 and 5).

The target area selected for this study is the drainage basin of the city of Asheville in North Carolina (Fig. 3). This watershed has a surface area of approximately $2,400\,\text{km}^2$ ($930\,\text{mi}^2$) and contains 88 nodes of the model's inner domain. It lies within the
region of influence of TCs (Zurndorfer et al., 1986). In fact, both TCs making landfall along the Gulf Coast and TCs making landfall along the Atlantic Coast can affect this watershed.

As an illustration of the transposition method, Fig. 4 shows the transposition of the surface zonal wind velocity in Hurricane Ivan for an amount of shift of $0.95°$ E and $4.10°$ N. The predominance of the red and green colors in the top most plot shows



that the storm is initially embedded within the trade winds[4]. As expected for a TC, the perturbation zonal wind field clearly exhibits a dipolar nature as can be seen in the right and bottom-right plots in Fig. 4.

Hurricane Ivan was transposed in a direction orthogonal to its direction of propagation at the simulation start date (Fig. 5). The transposition exercise was first performed for 29 increments of shift (including zero shift), from 1.67° W and 7.18° S to

1.67° E and 7.18° N, which corresponds to the black dots in Fig. 5. The WRF model was run for each of these increments of shift, and the maximum 72-hour (3-day) accumulated precipitation over the target watershed, which corresponds to the 72-h time window that contains the largest basin average precipitation depth, was calculated for every simulation. Results for this first step are presented in Fig. 6a. Note that the shifting results are represented by plotting them only against the West-East component of the shift that occurs along the line of black dots shown in Fig. 5. As such, the $x$-axis of Fig. 6a ranges from

1.67° W to 1.67° E. From this figure, it is observed that as the amount of shift increases from 1.67° W and 7.18° S, the 72-h basin average precipitation depth suddenly increases from about 2 mm (0.079 in) for an amount of shift of 1.19° W and 5.13° S to 310 mm (12.2 in) for an amount of shift of 1.07° W and 4.61° S. The 72-h basin average precipitation depth remains larger than 100 mm (3.94 in) until the amount of shift is increased over 0.71° E and 3.08° N for which the 72-h basin average precipitation depth drops to about 11 mm (0.43 in).

Figure 6a shows that the shifting window from 1.07° W and 4.61° S to 0.60° E and 2.56° N corresponds to the amounts of shift for which the target watershed is affected by the intense part of the precipitation field spawned by Hurricane Ivan. The two largest 72-h basin average precipitation depths occurred for an amount of shift of 1.07° W and 4.61° S, for which it is 310 mm (12.2 in), and for an amount of shift of 0.24° W and 1.02° S, for which it is 234 mm (9.21 in).

The second step of the maximization procedure (through storm transposition) is to refine around the local maxima obtained

in the previous step. This refinement was performed by considering the increments of shift halfway between the local maxima and the neighboring increments of shift. The results for the two aforementioned maxima of the 72-h basin average precipitation depth are presented in Fig. 6b. The first refinement confirms that the first peak of the 72-h basin average precipitation occurring for an amount of shift of 1.07° W and 4.61° S is larger than the peak associated with an amount of shift of 0.24° W and 1.02° S. As a result, the last refinement presented in Fig. 6c is performed only around the first peak. As many refinement steps as

necessary can be carried out. In the case of the transposition of Hurricane Ivan, Fig. 6 shows that the maximum 72-h basin average precipitation depth does not change appreciably from the second refinement step to the third refinement step. As a result, an estimation of the maximum 72-h basin average precipitation depth that Hurricane Ivan could have caused over the Asheville watershed (if it had passed over this area) is given by the red diamond in Fig. 6c and it is equal to approximately 348 mm (13.7 in).

Figure 7 shows that Hurricane Ivan responds nonlinearly to the transposition of its initial conditions. Indeed the location of the precipitation field does not change homogeously as the amount of shift is increased from 1.67° W and 7.18° S to 1.67° E and 7.18° N. For example, the 72-h accumulated precipitation field corresponding to an amount of shift of 1.07° W and 4.61° S (first plot on Row 3 in Fig. 7) is located east of the 72-h accumulated percipitation field corresponding to an amount of shift of 1.01° W and 4.36° S (second plot on Row 3 in Fig. 7). This behavior explains the presence of multiple peaks in the graphs of

---

[4]The trade winds are prevailing easterly winds that circle the Earth near the equator.



the 72-h basin average precipitation depth as a function of the zonal component of the shift presented in Fig. 6. This nonlinear response of the TC's track to a change in location of the initial vortex is even more striking in the case of Hurricane Frances presented in Section 4.

Figure 7 also shows that the simulated precipitation field in the case of zero shift (first plot on Row 6) is located east of the observed precipitation field (Fig. 8c). Given 1) the strong nonlinearity involved in the dynamics of a TC, 2) the fact that we used no nudging and data assimilation, and 3) the early simulation start date (about ten days before the time of landfall), it is not expected that the numerical model manages to reproduce accurately the track of the TC, including the time and location of landfall. Therefore, in order to place the simulated precipitation field in the right location, it is necessary to use a later simulation start date, as was done for the calibration of the WRF model discussed in the appendix for which the simulation start date was only two days before the time of landfall.

Figure 8 compares the 7-day accumulated precipitation field (from 09/14 00:00 UTC until 09/21 00:00 UTC) for the simulation which maximized the 72-h basin average precipitation depth to the observed 7-day accumulated precipitation field obtained from the National Center for Environmental Prediction (NCEP) Stage IV precipitation dataset (Lin and Mitchell, 2005)[5]. It is observed that the maximized precipitation field is overall significantly more intense than the observed field, which shows that the physically based transposition method does not result in a simple transposition of the storm's precipitation field, as it is often assumed in the traditional PMP approaches.

In order to explain the difference in intensity between the maximized precipitation field and the observed precipitation field, we calculated the time-averaged integrated vapor transport (IVT) field and its divergence for 1) the storm resulting from the maximization of the 72-h basin average precipitation depth (corresponding to Fig. 8b) and 2) the storm resulting from the calibration of the WRF model (corresponding to Fig. A1c). Given the good agreement between the observed precipitation field (Fig. 8c) and the precipitation field of the storm resulting from the calibration (Fig. A1c), the simulated moisture transport field (in calibration) is expected to be close to the moisture transport field of the original (i.e. observed) storm.

The IVT field (or moisture advection field) is a 2-dimensional vector field which quantifies the horizontal transport of water vapor by the wind. It is given by the following relationship:

$$IVT = \int_{z=0}^{z_{top}} \rho_v \mathbf{U} dz = \int_{z=0}^{z_{top}} q_v \rho \mathbf{U} dz = \frac{1}{g} \int_{p=0}^{p_{surf}} q_v \mathbf{U} dp \tag{1}$$

where $z$ is the vertical coordinate (height), $z_{top}$ is the height of the top of the atmosphere, $\rho_v$ is the density of the water vapor, $q_v$ is the water vapor mixing ratio, $\rho$ is the density of dry air, $p$ is pressure, $p_{surf}$ is the surface pressure, $g$ is the gravitational acceleration, and $\mathbf{U}$ is the wind velocity vector.

Results are presented in Fig. 9. The vector field shows the IVT averaged over the period from 09/14 00:00 UTC until 09/21 00:00 UTC. The associated color plot gives the divergence of the time-averaged IVT field. Positive values indicate a decrease of the mass of water vapor contained in an atmospheric column whereas negative values indicate an increase in the mass of

---

[5]Stage IV is a NCEP-generated mosaic of regional multi-sensor precipitation analysis produced by National Weather Service River Forecast Centers (RFCs) since 2002.





water vapor contained in an atmospheric column. Comparing Fig. 9a with Fig. 8b on the one hand, and Fig. 9b with Fig. A1c on the other hand, it is obvious that the regions of intense precipitation coincide with the regions where the convergence (the negative of the divergence) of the IVT is maximized. As a result, the local increase of water vapor in the atmosphere through convergence of the IVT is likely to have played a major role in the generation of intense precipitation in Hurricane Ivan.

5    Interestingly, the magnitude of the IVT in the storm resulting from the maximization is only slightly larger than the magnitude of the IVT in the storm resulting from the calibration. In particular, around the target area and downstream from the target area, both magnitudes are approximately the same. As a consequence, the significant increase in the intensity of the precipitation field in Hurricane Ivan between the maximized case and the calibrated case seems to be due to an increase of the convergence of the IVT rather than to an increase of the magnitude of the IVT.

## 10    4    Transposition of Hurricanes Floyd (1999), Frances (2004) and Isaac (2012)

In this section, we apply the transposition method to three other TCs that generated torrential precipitation in the United States: Hurricane Floyd (1999), Hurricane Frances (2004), and Hurricane Isaac (2012). The WRF model's options and the simulation start dates used for the simulations of these TCs are given in Table 2. They were selected based on the calibration of the model discussed in the appendix. Other modeling choices (time step, number of vertical layers, etc.) are the same as for Hurricane

Ivan as discussed at the beginning of Section 3. Initial and boundary conditions are from CFSR. The nested domains used for the simulation of Hurricane Isaac are the same as for Hurricane Ivan. In the case of Hurricanes Floyd and Frances, the intermediate and inner domains were taken slightly more east in order to account for the location of the precipitation fields in the original storms.

Figure 10 shows the location of the center of low mean sea level pressure before transposition (corresponding to the green

points) and after transposition (corresponding to the black points). In the case of Hurricane Ivan, Fig. 7 shows that the precipitation field for zero shift (first plot on Row 6) already goes through the target watershed. This is the reason why we considered the amounts of shift symmetrically around zero for Hurricane Ivan (Fig. 5). In the case of Hurricane Floyd, the precipitation field for zero shift is located significantly east of the target watershed (not shown), which explains why only negative amounts of shift (westerly and southerly) were considered for Hurricane Floyd (Fig. 10a). In the case of Hurricane Frances, the precip-

itation field corresponding to zero shift is located slightly west of the target watershed (not shown), which explains why more positive (easterly and northerly) than negative (westerly and southerly) amounts of shift were considered (Fig. 10b). Finally, in the case of Hurricane Isaac, the precipitation field corresponding to zero shift is located significantly west of the target watershed (not shown), which explains why positive amounts of shift (easterly and northerly) were predominantly considered (Fig. 10c).

The results for the transposition of these three hurricanes are presented in Fig. 11. Figure 11 shows the 72-h basin average precipitation depth as a function of the zonal component of the shift. The y-axis represents the precipitation depth of the 72-h time window for which the precipitation over the target is the largest for each simulation.

It is observed that the results for the transposition of Hurricanes Floyd and Isaac (Fig. 11a and c) are similar to the results





for the transposition of Hurricane Ivan to the extent that the graphs of the 72-h basin average precipitation depth as a function of the zonal component of the shift contain well defined peaks, and the maximum 72-h precipitation depths (given by the red diamonds) were obtained through the refinement steps described in Section 3. However, the results are significantly different in the case of Hurricane Frances (Fig. 11b). Indeed the graph of the 72-h basin average precipitation depth as a function of the

5 zonal component of the shift is very oscillatory. Actually, the refinement procedure failed in the case of Hurricane Frances. As refinement steps were performed, new peaks kept appearing in the graph and the existing peaks got narrower. As a result, it was necessary to use a much finer shifting increment for the whole shifting window from 0.077° W and 0.48° S to 0.65° E and 4.0° N in order to obtain Fig. 11b.

The simulated track of Hurricane Frances was observed to be extremely sensitive to the location of the initial vortex. For

example, Fig. 12 shows the 7-day accumulated precipitation field (from 09/04/2004 00:00 UTC until 09/11/2004 00:00 UTC) and the track of the storm (given by the black dots) for the simulation which maximized the 72-h basin average precipitation depth (Fig. 12b), and for the simulations associated with a shifting amount slightly more south-west (Fig. 12a), and slightly more north-east (Fig. 12c) than for the maximized case. More precisely, the shifting amount for Fig. 12a is 0.066° W and 0.41° S, whereas it is 0.055° W and 0.34° S for Fig. 12b, and 0.044° W and 0.27° S for Fig. 12c. As a result, only about 0.07°

(∼ 8 km) separates the initial vortex in the simulations corresponding to Fig. 12a and Fig. 12c from the initial vortex in the simulation corresponding to Fig. 12b. Yet, it is striking to see how different the tracks are. The locations of landfall differ by more than 100 km between one case and another, causing the intense precipitation from Hurricane Frances to affect different regions. Furthermore, the track does not respond homogenously to the amount of shift of the initial vortex. Indeed, it would be expected for the track in Fig. 12c to be located east of the track in Fig. 12b since the initial vortex for the simulation corre-

sponding to Fig. 12b is located more west and south than the initial vortex for the simulation corresponding to Fig. 12c. Not only the track of Hurricane Frances in Fig. 12c is located west of the track in Fig. 12b, but also it is located west of the track in Fig. 12a, for which the initial vortex is even more west and south. Actually, the three tracks remain close to each other until Hurricane Frances approaches southeastern Florida (not shown). As Frances gets closer to southeastern Florida, and especially after landfall, a dramatic change is observed in the behavior of the track of the storm, leading to the results of Fig. 12.

Figures 13, 15 and 16 present the precipitation fields (in the inner domain) for the simulations which maximized the 72-h basin average precipitation depth (corresponding to the red diamonds in Fig. 11) along with the observed precipitation fields for Hurricanes Floyd, Frances, and Isaac, respectively. In the case of Hurricanes Frances and Isaac, the observed precipitation fields were obtained from the NCEP Stage IV dataset, whereas the observed precipitation field for Hurricane Floyd (which occurred before the period of availability of Stage IV starting in 2002) was obtained from the National Oceanic and Atmospheric

Administration (NOAA)[6] (Fig. 14).

Figure 15 shows that, for Hurricane Frances, the maximized precipitation field is overall significantly more intense than the observed precipitation field, as was the case for Hurricane Ivan. However, in the case of Hurricane Isaac (Fig. 16), the maximized precipitation field is overall as intense as the observed precipitation field, whereas in the case of Hurricane Floyd (Fig. 13 and 14) the maximized precipitation field is overall slightly less intense than the observed precipitation field. These results

---

[6]http://www.wpc.ncep.noaa.gov/tropical/rain/floyd1999.html





confirm that the new transposition method does not lead to a simple transposition of the observed precipitation field over the target area. The intensity and structure of the transposed precipitation field depend on the new track of the TC, and on how the transposed TC interacts with its environment including the local topography and the presence of other synoptic and mesoscale systems. These interactions are explicitly accounted for by the RAM which crucially conserves the mass, momentum, and 5 energy.

The maximum 72-h basin average precipitation depth over the Asheville watershed obtained in this study from the maximization (through transposition) of the precipitation from four hurricanes is equal to 427 mm (16.8 in). It resulted from the transposition of Hurricane Frances. Interestingly, this amount compares favorably with estimates obtained using the traditional approach. Zurndorfer et al. (1986) provided estimates of 1- to 72-h PMP and Tennessee Valley Authority (TVA) precipitation 10 for basins ranging between 5 and 3,000 mi$^2$ in the Tennessee Valley watershed. The TVA precipitation was defined as "the level of precipitation resulting from transposition and adjustment (without maximization) of outstanding storms". In this case, "maximization" refers to the moisture maximization step as discussed in the introduction. Their estimate of the 72-h TVA precipitation for the drainage basin of the city of Asheville was 503 mm (19.8 in), which is relatively close to the maximum 72-h precipitation depth obtained from the numerical atmospheric model-based transposition of Hurricane Frances. However, 15 their estimate for the 72-h PMP was 869 mm (34.2 in), which is about twice as large as the maximum 72-h precipitation depth obtained from the physically based transposition of Hurricane Frances.

## 5   Procedure for the physically based estimation of the PMP through numerical transposition of TCs

In the previous sections, a method to maximize the precipitation from a TC over a specified target area was presented and applied to four hurricanes that spawned torrential precipitation in the United States. The drainage basin of the city of Asheville 20 was used as the target area. In this section, we propose a procedure to estimate the PMP for a target area whose intense precipitation is caused by TCs, as it is usually the case for a sufficiently large[7] watershed in the eastern United States.

The first step is to identify the TCs that can potentially generate intense precipitation over the target area. It was shown previously that the track of a TC may be very sensitive to small changes in the location of the storm at the simulation start date, and that the response of the track to such changes may be highly nonlinear. As a result, a TC that affected a region far 25 from the target may still be able to produce significant precipitation over the target after transposition. Similarly, a TC that did not originally make landfall may be brought over land and pass over the target after transposition. Thus most historical TCs should be considered, except for those for which it is obvious that a small shift of the initial vortex will not bring the TC over the target area. Let us write $N_{TC}$ as the number of TCs selected in the first step.

Second, for each of these $N_{TC}$ TCs, it is necessary to calibrate the WRF model as discussed in the appendix in order to 30 identify an appropriate set of the model's physics parameterization options. In this article, we considered one set of options for each storm (Tables 1 and 2). However, several sets of options may provide satisfactory calibration results. If this is the case, one should consider all the sets of the model's options that give satisfactory calibration results for a given TC in order to account

---

[7]$> 100$ mi$^2$ according to Zurndorfer et al. (1986)





for the model uncertainties. Let us write $N_{so}(i)$ as the number of sets of options retained for the $i^{th}$ TC.

Ideally, the choice of the simulation start date should respect the restrictions emphasized in Section 2 including the facts that the TC in the initial conditions should be far enough from land, it should be intense enough, and it should not have started its extratropical transition. In this article, one simulation start date was considered for each event. For example, in the case of

Hurricane Ivan, the simulation start date was 09/06/2004 00:00 UTC. However, in order to account for the uncertainties related to the initial conditions in the estimation of the PMP, one should consider as many simulation start dates as possible, making sure that they (ideally) satisfy the aforementioned restrictions. For example, in the case of Hurricane Ivan, since the CFSR data used for initial and boundary conditions is provided with a temporal resolution of 6 hours, one may also consider the following simulation start dates: 09/05/2004 18:00 UTC, 09/06/2004 06:00 UTC, 09/06/2004 12:00 UTC, 09/06/2004 18:00 UTC, etc.

Let us write $N_{ic}(i)$ as the number of simulation start dates retained for the $i^{th}$ TC. Different reanalysis atmospheric datasets may also be used to account for uncertainties related to the initial and boundary conditions.

An estimate of the PMP over the target area is obtained by applying the transposition method presented in this article to all the identified TCs and for all the corresponding sets of options and simulation start dates. This amounts to performing the transposition exercise $N$ times where $N = \sum_{i=1}^{N_{TC}} N_{so}(i) \times N_{ic}(i)$. This corresponds to a significant computational effort.

However, with today's technology, such a computational effort is feasible. Several recent studies reported on the application of the dynamical downscaling technique to the PMP estimation problem and other hydrological problems. For example, Ishida et al. (2014) maximized the 72-h basin average precipitation depth for 61 atmospheric rivers over three watersheds in California using the atmospheric boundary condition shifting method discussed in the introduction. This amounted to thousands of simulations performed with the fifth Generation Mesoscale Atmospheric Model (MM5) (Grell et al., 1994) with 4-level nested

domains and spatial resolution from 81 km in the outer domain to 3 km in the inner domain. Trinh et al. (2016) simulated future flow conditions in the Cache Creek watershed in California over the $21^{st}$ century. To achieve this objective, they dynamically downscaled to 3-km resolution thirteen climate projections from two GCMs under four emission scenarios and several initial conditions, and used these atmospheric state variables as input to the WEHY model, a physically based hydrologic model based on upscaled conservation equations (Chen et al., 2004a, b; Kavvas et al., 2004).

Moreover, in this study, the shifting increments used in the transposition of the four hurricanes were relatively small. For example, 29 increments of shift were first considered for Hurricane Ivan (Fig. 5). In practice, many TCs can be ruled out by performing the transposition exercise with a larger shifting increment. For example one may first consider five increments of shift and check if the TC gets closer to the target. If this is not the case, this TC can be disregarded for the purpose of PMP estimation.

In the end, one obtains $N$ realizations of a TC spawning intense precipitation over the target area. The PMP may be chosen as the largest 72-h (or other duration) basin average precipitation depth among these N realizations. Besides, the proposed procedure offers a way to quantify the uncertainties associated with the PMP estimate due the uncertainties in the model and in the initial conditions. Suppose that the maximum precipitation depth over the target is obtained for the $j^{th}$ TC. In this case, one may investigate the $N_{so}(j) \times N_{ic}(j)$ realizations of this storm in order to analyze how sensitive the maximized precipitation

depth is to the model's options and to the initial environment.





Finally, we note that the proposed procedure for the estimation of the PMP can handle non-stationarity in the hydroclimate, which is not the case for the traditional approaches. Indeed, since it is physically based, the new transposition method can be applied to future TCs, through dynamical downscaling of projections from GCMs, which would allow investigating how the PMP evolves as the climate changes.

## 6   Conclusions

In this article, a new storm transposition method designed for the transposition of TCs was presented. This method is fully physically based, as it uses a RAM to numerically simulate a TC and its precipitation field. As a result, it has the fundamental advantage of conserving the mass, momentum, and energy in the system since the RAM numerically solves the equations governing the conservation of these quantities. The new storm transposition method is based on the shifting of the initial vortex
of the TC at the simulation start date. More precisely, the TC in the initial conditions is first separated from its background environment, then shifted, and finally recombined with the background environment. The objective of this method is to find the amount of shift which maximizes the precipitation depth over a given target area.

In this study, the transposition method was applied to four hurricanes that had spawned torrential precipitation in the United States, namely Hurricanes Floyd (1999), Frances (2004), Ivan (2004), and Isaac (2012). The drainage basin of the city of
15 Asheville was selected as the target. It was found that the tracks of these TCs are generally very sensitive to changes in the location of the initial vortex, and that the response of the tracks to such changes is nonlinear. In particular, a small shift of the initial vortex can result in a significant change in the location, structure, and intensity of the precipitation field, thus putting into question both the legitimacy of the conventional transposition of the precipitation field from a TC (as is often the case in the traditional PMP approaches) and the existence of a meteorologically homogeneous region that sets the transposition limits. The
20 precipitation fields resulting from the numerical atmospheric model-based maximization (through transposition) of Hurricanes Frances and Ivan are overall significantly more intense than the observed precipitation fields. The investigation of the IVT and its convergence in the case of Hurricane Ivan revealed that the increased intensity in the maximized precipitation field is due to an increase in the convergence of the IVT rather than to an increase in the magnitude of the IVT. In the case of Hurricane Floyd, the numerical atmospheric model maximized precipitation field was overall slightly less intense than the observed precipitation
field, whereas for Hurricane Isaac, the maximized precipitation field was overall as intense as the observed precipitation field.

The maximum 72-h basin average precipitation depth resulting from the transposition of the four hurricanes was equal to 427 mm (16.8 in). It compares favorably to the 72-h TVA precipitation obtained by the generalized method (Zurndorfer et al., 1986) which is equal to 503 mm (19.8 in), but remains about half of the 72-h PMP obtained with the generalized method which is equal to 869 mm (34.2 in). Finally, a procedure is proposed for the physically based estimation of the PMP over a given
target area based on the new transposition method. If possible, the transposition exercise should be performed using different sets of the RAM's options, and different simulation start dates, in order to quantify the uncertainties in the PMP estimate due to the model uncertainties and uncertainties in the initial conditions.





## Appendix A:  Calibration of the WRF model

This appendix details the calibration of the WRF model for the reconstruction of the four TCs. These calibration results are from Mure-Ravaud et al. (2017a) and Mure-Ravaud et al. (2017b) (unpublished manuscript). In these studies, thirteen intense TCs that affected the Eastern United States during 1999-2016 were reconstructed by the WRF model, including the four hur-

ricanes investigated in this paper: Hurricanes Floyd, Frances, Ivan and Isaac.

For each event, the authors tried several combinations of the WRF model's physics parameterization options, which include the microphysics options, cumulus options, planetary boundary layer options, shortwave and longwave radiation options, surface layer options, and land surface options. In each case, the "best" combination, given in Table 1 for Hurricane Ivan and in Table 2 for Hurricanes Floyd, Frances and Isaac, was selected based on the model's performance in reproducing the observed

precipitation field. The calibration was performed by changing the parameterization schemes starting from a list of combinations of these schemes based on the literature and on the authors' experience in the numerical modeling of storm systems. It was terminated when satisfactory results were obtained in terms of the reconstruction of the TCs' precipitation fields.

The performance of the model in reproducing the precipitation fields was assessed in two ways: first by the use of three metrics (described below), and second by visual examination of the plots of the simulated and observed precipitation fields.

Since Hurricane Floyd (1999) occurred before the period of availability of the Stage IV dataset, the metrics were not calculated for this event. What was determined to be the "best" result was often a compromise between the quantitative results in terms of three metrics, and the more qualitative visual examination of the accumulated precipitation fields.

The first quantitative metric was the relative error in the inner-domain-averaged total precipitation:

$$\frac{P_{T_{sim}} - P_{T_{obs}}}{P_{T_{obs}}} \tag{A1}$$

where $P_{T_{sim}}$ is computed by accumulating simulated precipitation for a given time period at each cell which was then averaged over the inner domain, and $P_{T_{obs}}$ is defined similarly for the observed precipitation. The relative error in inner-domain-averaged total precipitation indicates if the model adequately simulated the total precipitation amount over the period of interest.

The second quantitative metric utilized was the overlap percentage, computed for several precipitation thresholds. The

overlap percentage is equal to 100 times the number of grid points where both the observation and the simulation are above the threshold, divided by the number of points where the observation is above a given threshold. The overlap percentage indicates whether the model could place the storm system in the appropriate location. Six precipitation thresholds corresponding to percentiles for the observed precipitation intensity: $50^{th}$ percentile, $75^{th}$ percentile, $90^{th}$ percentile, $95^{th}$ percentile, $97.50^{th}$ percentile, and $99^{th}$ percentile thresholds were used.

The third quantitative metric was the area ratio between the simulated and observed precipitation fields (precipitation field area ratio; PFAR) for each threshold. If this ratio is less than one, the model underestimated the size of the precipitation field above the threshold. If the ratio is larger than one, the model overestimated the size of the precipitation field above the threshold. As a consequence, the best performance is obtained when the overlap percentage is close to 100% and the PFAR is close to 1.

Figure A1 presents the simulated precipitation fields in the four hurricanes. These simulated precipitation fields can be





compared to the observed precipitation fields shown in Fig. 8c, 14, 15c and 16c. Table A1 summarizes the calibration results in terms of the three metrics for Hurricanes Frances, Ivan and Isaac.

Figure A1 and Table A1 show that the WRF model provided overall good performance in reconstructing the precipitation fields associated with the four TCs.

5 *Competing interests.* The authors declare that they have no conflict of interest.

*Acknowledgements.* This work was supported by the U.S. Nuclear Regulatory Commission Office of Nuclear Regulatory Research in the form of a Coordinated Grant via the U.S. Geological Survey State Water Resources Research Institutes Program (USGS Award No. G15AP00045; NRC Interagency Agreement No. NRC-HQ-60-14-I-0025).




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





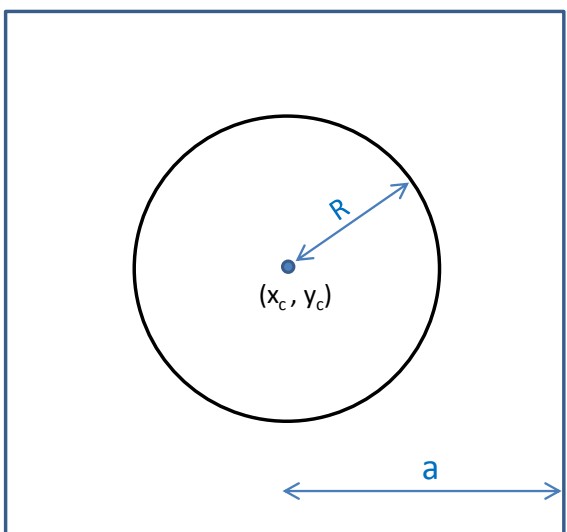

**Figure 1.** Configuration used for the interpolation of the background atmospheric fields. The black circle of center $(x_c, y_c)$ and of radius $R$ is the region of influence of the TC. The blue square of side $2a$ is the region used to reconstruct the background fields inside the circle through interpolation.




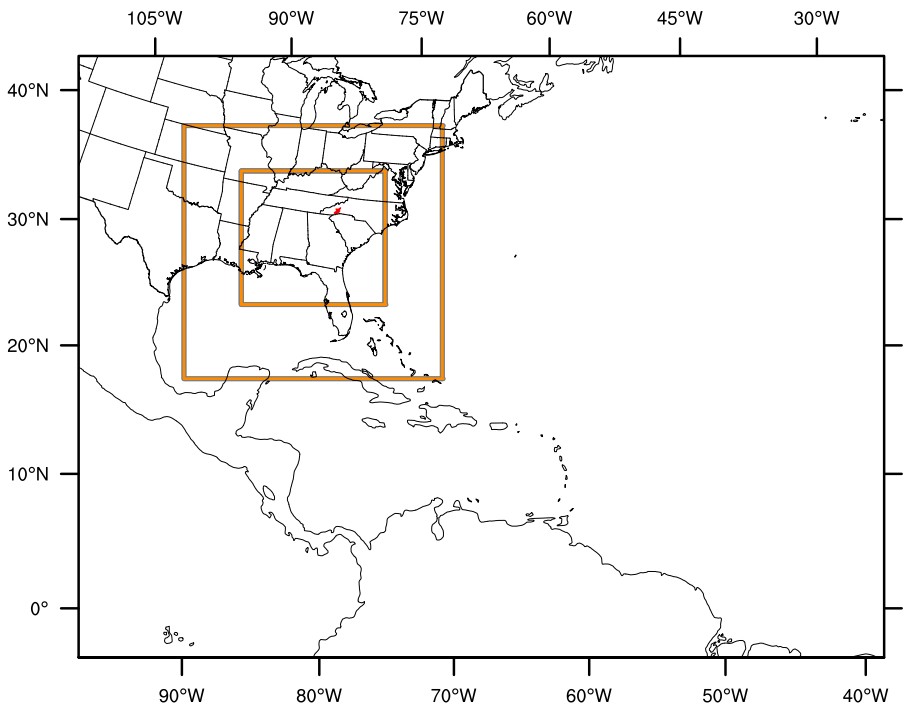

**Figure 2.** Nested domains used for the simulations of Hurricane Ivan. The small red area in western North Carolina is the target watershed, presented in Fig. 3.





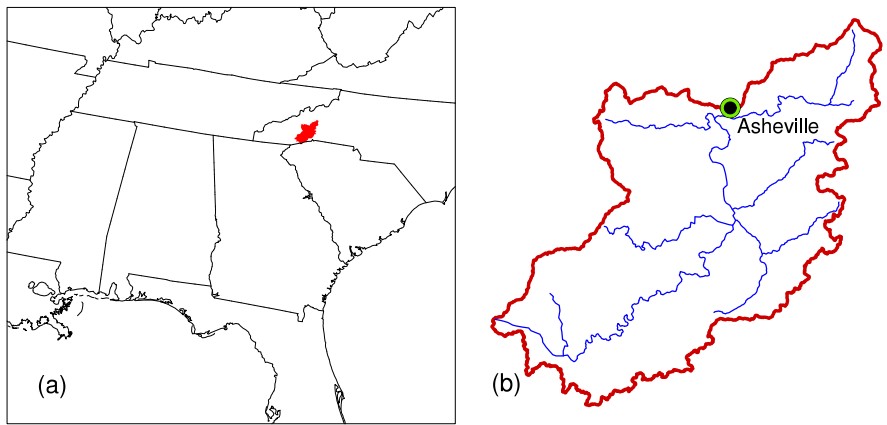

**Figure 3.** Target area used for the transposition. (a) The target area is shown in red within the model's inner domain. (b) The target area corresponds to the drainage basin of the city of Asheville, N.C.





**Figure 4.** Application of the transposition procedure to the initial surface zonal wind velocity $(\mathrm{m\,s^{-1}})$ in Hurricane Ivan (2004).





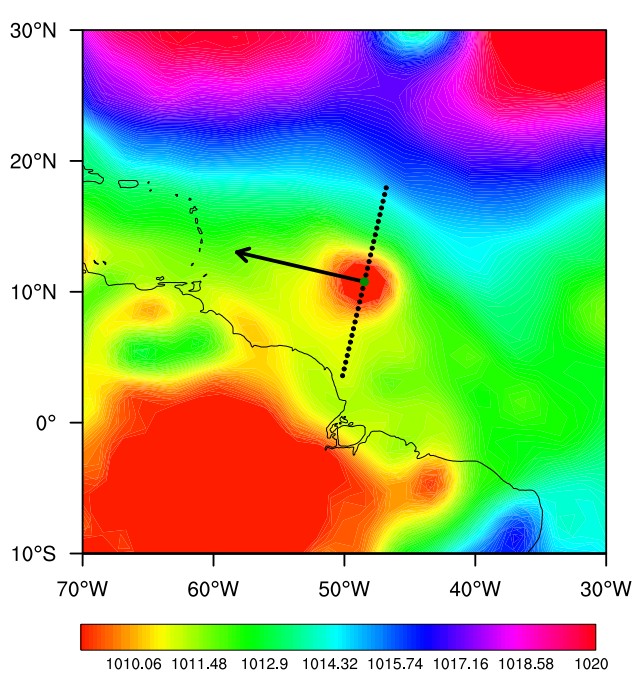

**Figure 5.** The color plot shows the mean sea level pressure field (mbar) on 09/06/2014 00:00 UTC (from CFSR) for zero shift. The green point shows the location of the center of low pressure in the original TC (zero shift). The black points show the location of the center of low pressure after shifting. The black arrow indicates the directon of propagation of Hurricane Ivan.





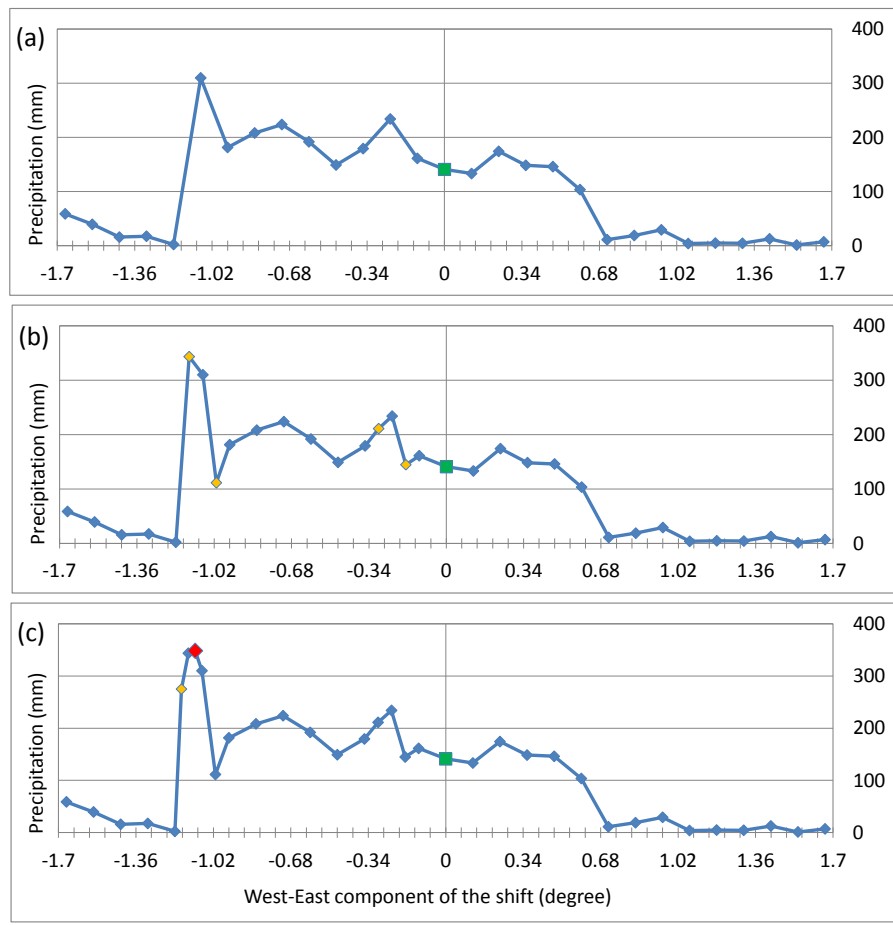

**Figure 6.** 72-h basin average precipitation depth from Hurricane Ivan as a function of the West-East component of the shift along the transect shown in Fig. 5. (a) Results for the 29 increments of shift first considered (Fig. 5). (b) Results after the first refinement. (c) Results after the second refinement. The green square gives the 72-h basin average precipitation depth in the case of no shift. The yellow diamonds show the refinement performed around the local maxima. The red diamond in (c) indicates the maximum 72-h basin average precipitation depth.





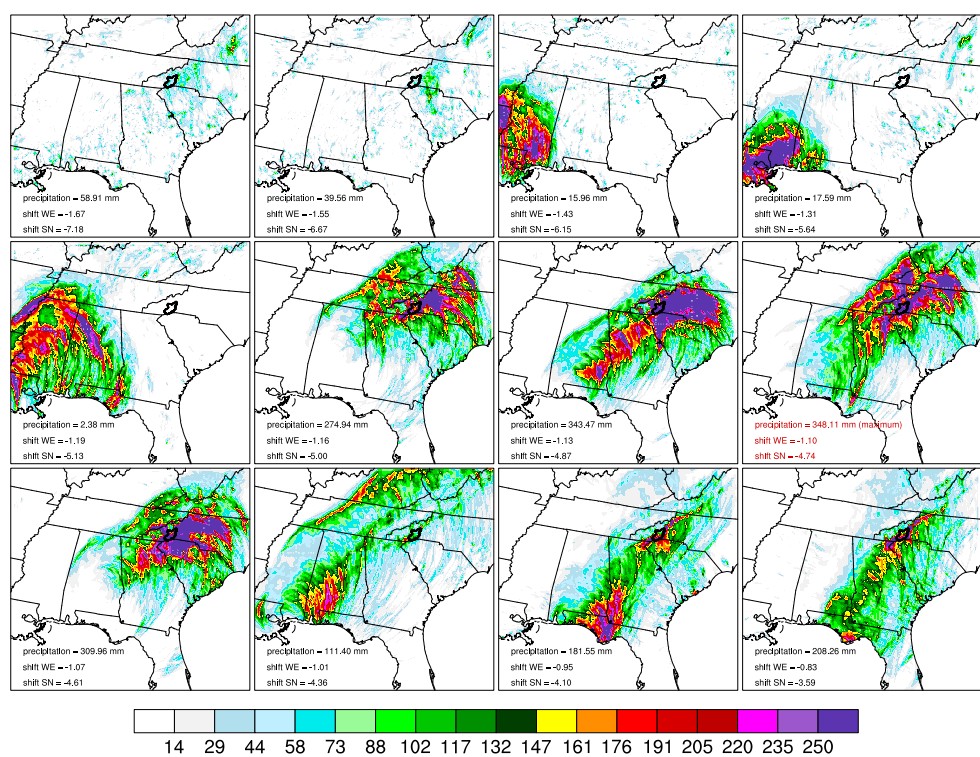

**Figure 7.** 72-h accumulated precipitation depth (mm) field in Hurricane Ivan as a function of the amount of shift. The first plot (top-left) corresponds to the most westerly and southerly shift (1.67° W and 7.18° S) while the last plot (bottom-right) corresponds to the most easterly and northerly shift (1.67° E and 7.18° N). The maximum 72-h basin average precipitation depth is obtained for the 8th plot (fourth plot on Row 2).





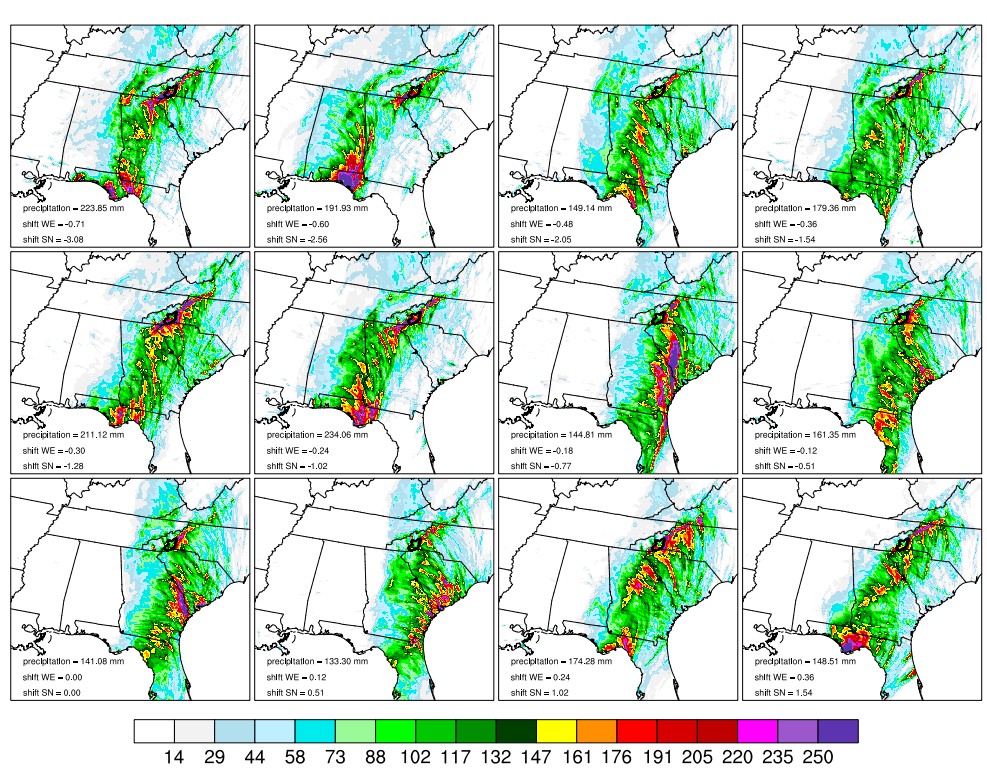

**Figure 7.** continued





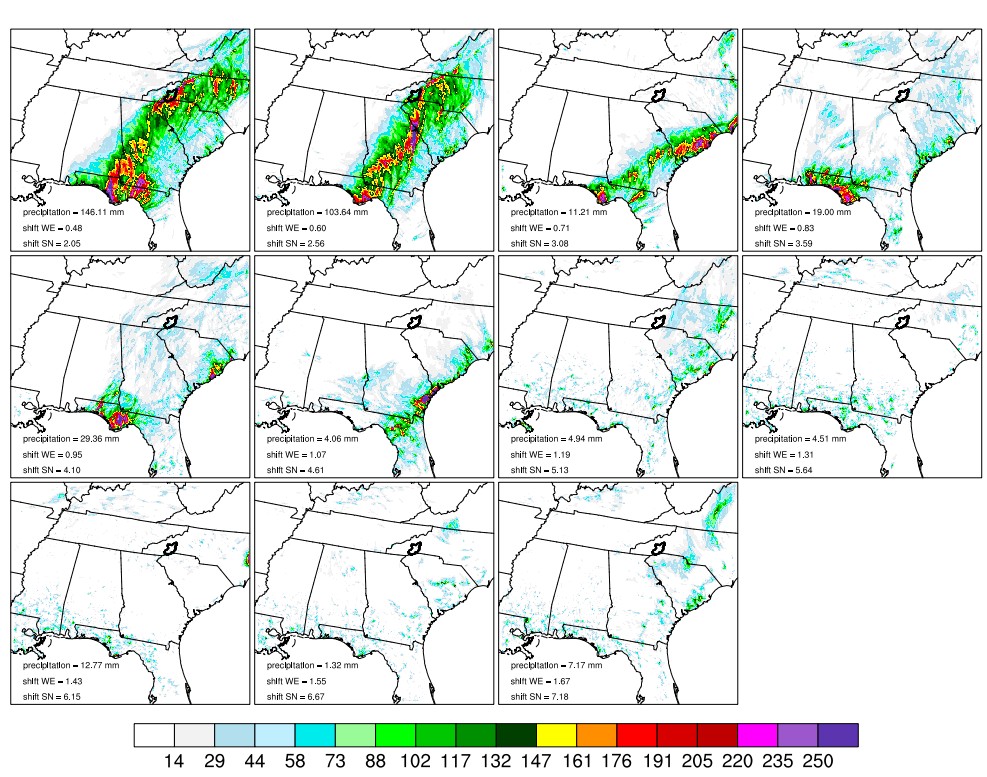

**Figure 7.** continued





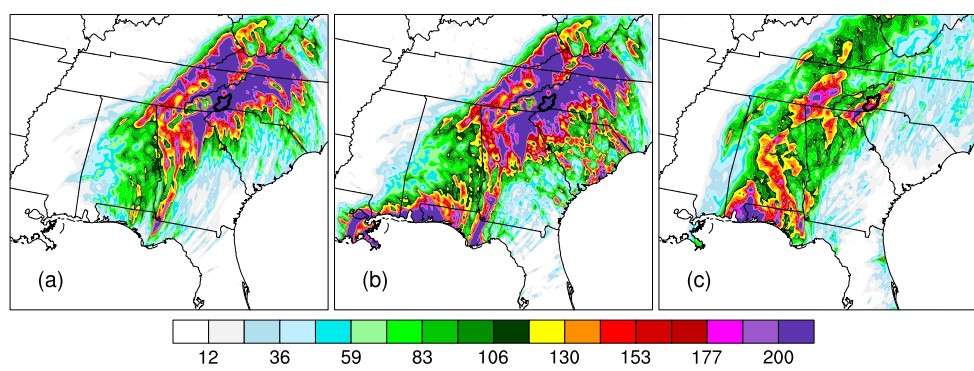

**Figure 8.** (a) 72-h accumulated precipitation depth (mm) field (from 09/16 08:00 UTC until 09/19 08:00 UTC) in Hurricane Ivan for the simulation which maximized the 72-h basin average precipitation depth. (b) 7-day accumulated precipitation field (from 09/14 00:00 UTC until 09/21 00:00 UTC) for the simulation which maximized the 72-h basin average precipitation depth. (c) Observed 7-day accumulated precipitation field (from 09/14 00:00 UTC until 09/21 00:00 UTC).

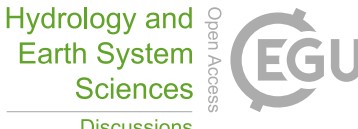

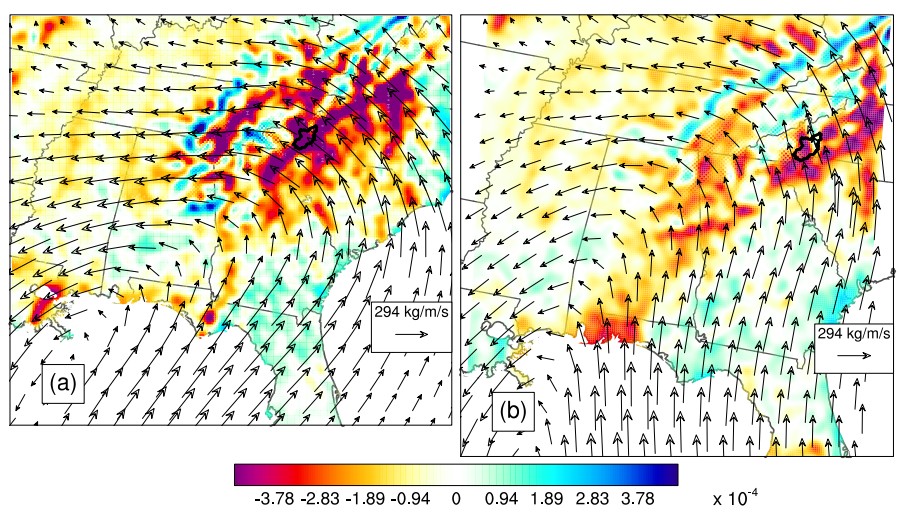

**Figure 9.** Arrow field: time-averaged (from 09/14 00:00 UTC to 09/21 00:00 UTC) integrated vapor transport ($\mathrm{kg\,m^{-1}\,s^{-1}}$). Color plot: Divergence of the time-averaged integrated vapor transport field ($\mathrm{kg\,m^{-2}\,s^{-1}}$) for (a) the simulation which maximized the 72-h basin average precipitation depth from Hurricane Ivan and (b) the simulation resulting from the calibration of the WRF model (see the appendix).



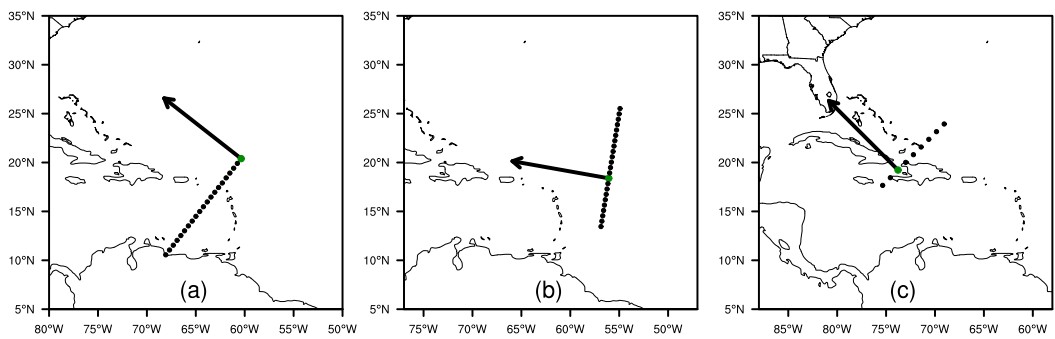

**Figure 10.** Location of the center of low mean sea level pressure in the initial conditions before shifting (green point) and after shifting (black points) for (a) Hurricane Floyd, (b) Hurricane Frances and (c) Hurricane Isaac. The black arrows show the direction of storm propagation.





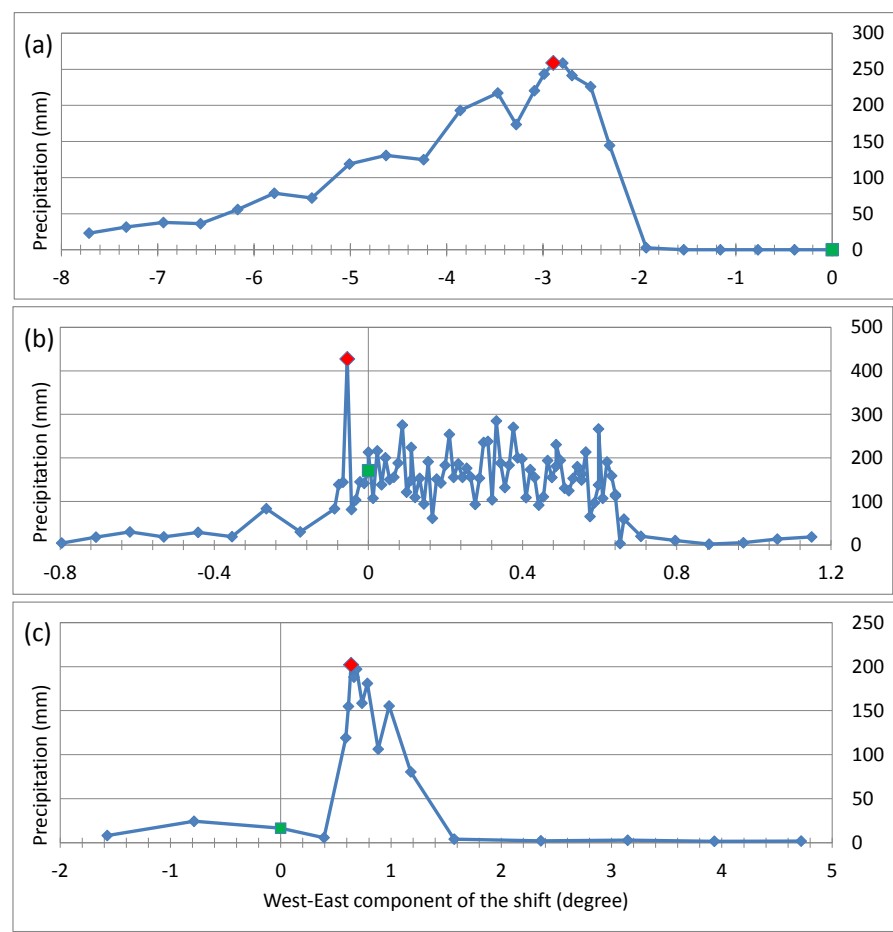

**Figure 11.** 72-h basin average precipitation depth as a function of the amount of shift along the transects shown in Fig. 10 for (a) Hurricane Floyd, (b) Hurricane Frances, and (c) Hurricane Isaac. The green squares give the 72-h basin average precipitation depth for zero shift. The red diamonds give the maximum 72-h basin average precipitation depth.





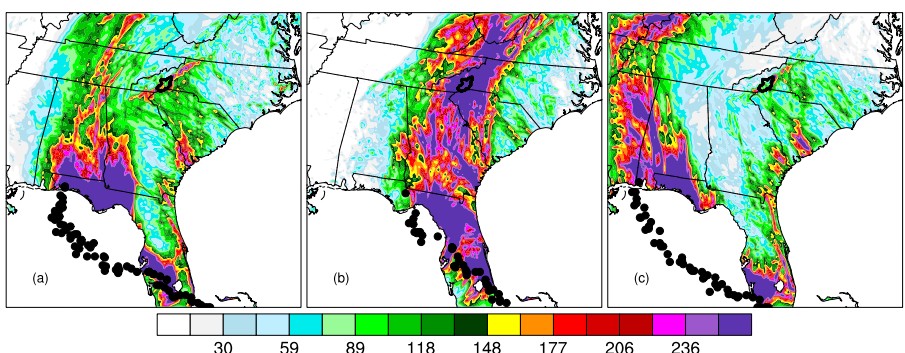

**Figure 12.** Illustration of the sensitivity of the track of Hurricane Frances to the location of the storm at the simulation start date. The color plot gives the 7-day (from 09/04/2004 00:00 UTC until 09/11/2004 00:00 UTC) accumulated precipitation (mm) field in Hurricane Frances for (a) the simulation associated with an amount of shift of $0.066°$ W and $0.41°$ S ; (b) the simulation which maximized the 72-h basin average precipitation depth, corresponding to an amount of shift of $0.055°$ W and $0.34°$ S; and (c) the simulation associated with an amount of shift of $0.044°$ W and $0.27°$ S. The black dots show the location of the center of low surface pressure with an hourly time increment from 09/04/2004 00:00 UTC until the time of second landfall.





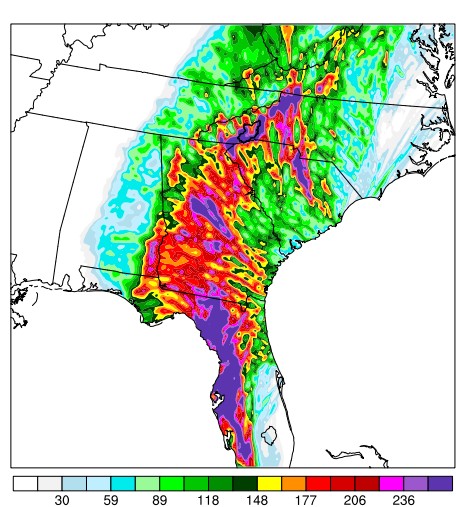

**Figure 13.** 72-h (from 09/15/1999 06:00 UTC until 09/18/1999 06:00 UTC) accumulated precipitation (mm) field in Hurricane Floyd for the simulation which maximized the 72-h basin average precipitation depth.



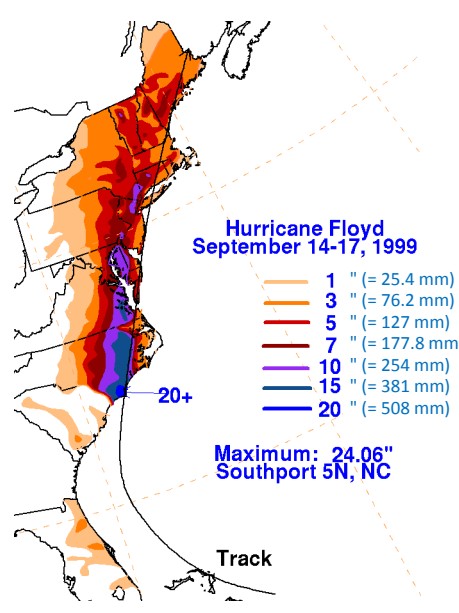

**Figure 14.** Observed 72-h (from 09/14/1999 00:00 UTC until 09/17/1999 00:00 UTC) accumulated precipitation field in Hurricane Floyd (adapted from NOAA; http://www.wpc.ncep.noaa.gov/tropical/rain/floyd1999.html)




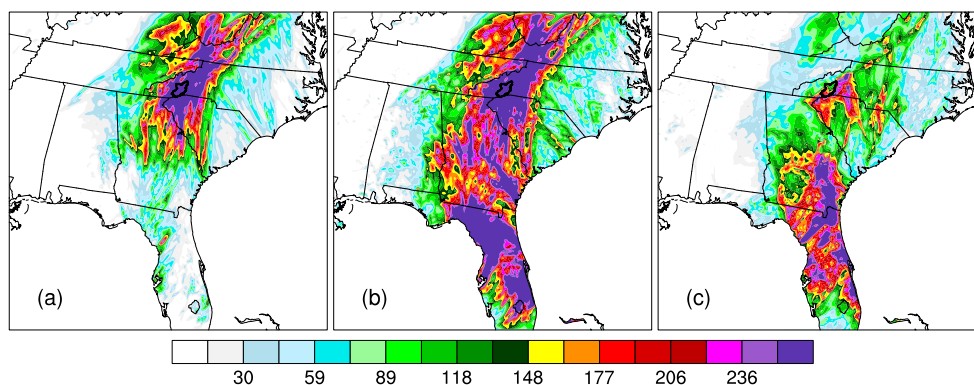

**Figure 15.** (a) 72-h (from 09/06/2004 23:00 UTC until 09/09/2004 23:00 UTC) accumulated precipitation (mm) field in Hurricane Frances for the simulation which maximized the 72-h basin average precipitation depth. (b) 7-day (from 09/04/2004 00:00 UTC until 09/11/2004 00:00 UTC) accumulated precipitation field in Hurricane Frances for the simulation which maximized the 72-h basin average precipitation depth. (c) Observed 7-day (from 09/04/2004 00:00 UTC until 09/11/2004 00:00 UTC) accumulated precipitation field in Hurricane Frances.





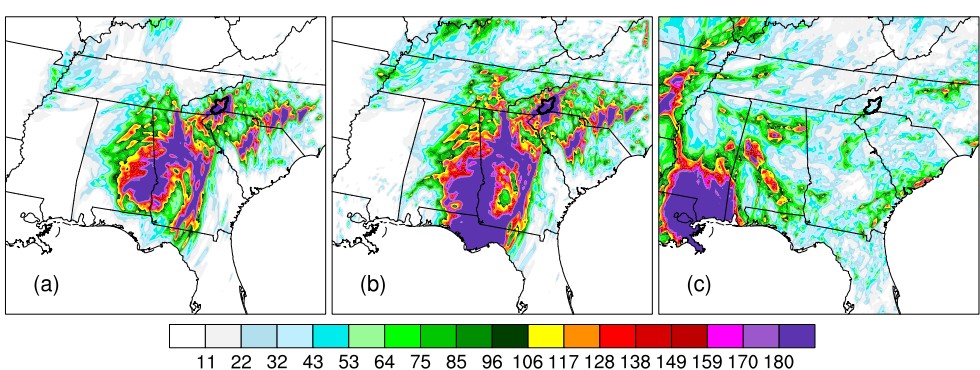

**Figure 16.** (a) 72-h (from 08/29/2012 12:00 UTC until 09/01/2012 12:00 UTC) accumulated precipitation (mm) field in Hurricane Isaac for the simulation which maximized the 72-h basin average precipitation depth. (b) 7-day (from 08/28/2012 12:00 UTC until 09/04/2012 12:00 UTC) accumulated precipitation field in Hurricane Isaac for the simulation which maximized the 72-h basin average precipitation depth. (c) Observed 7-day (from 08/28/2012 12:00 UTC until 09/04/2012 12:00 UTC) accumulated precipitation field in Hurricane Isaac.



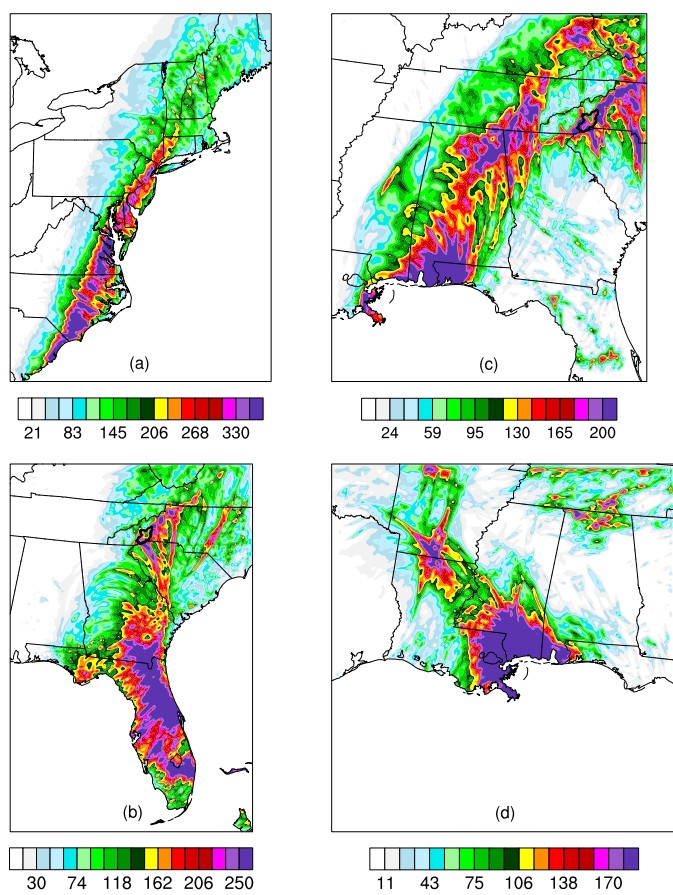

**Figure A1.** Reconstructed (in calibration) accumulated precipitation (mm) fields in the case of (a) Hurricane Floyd (09/15/1999 06:00 UTC to 09/18/1999 06:00 UTC); (b) Hurricane Frances (09/04/2004 00:00 UTC to 09/11/2004 00:00 UTC); (c) Hurricane Ivan (09/14/2004 00:00 UTC to 09/21/2004 00:00 UTC); Hurricane Isaac (08/28/2012 12:00UTC to 09/04/2012 12:00UTC). From Mure-Ravaud et al. (2017a) and Mure-Ravaud et al. (2017b) (unpublished manuscript).





**Table 1.** Parameterization schemes used for the simulation of Hurricane Ivan. The code number corresponding to each option according to the ARW Version 3 Modeling System User's Guide (Wang et al., 2016) is given between brackets.

| | |
|---|---|
| **Microphysics** | WRF Double Moment 6-class (WDM6) [16] |
| **Cumulus Parameterization** (domains 1 and 2 only) | New Simplified Arakawa-Schubert (SAS) [14] |
| **Planetary Boundary Layer** | Mellor-Yamada-Janjic (MYJ) [2] |
| **Longwave Radiation** | Rapid Radiative Transfer Model (RRTM) [1] |
| **Shortwave Radiation** | Dudhia [1] |
| **Land Surface** | Unified Noah land-surface model [2] |
| **Surface Layer** | Monin-Obukhov [2] |



**Table 2.** WRF options and simulation start dates used for the simulations of Hurricanes Floyd, Frances and Isaac. The code number corresponding to each option according to the ARW Version 3 Modeling System User's Guide (Wang et al., 2016) is given between brackets.

|  | Floyd | Frances | Isaac |
|---|---|---|---|
| **Microphysics** | WRF Double Moment 6-class (WDM6) [16] | Stony-Brook University 5 class (SBU-YLin) [13] | Lin (Purdue) [2] |
| **Cumulus Parameterization** (domains 1 and 2 only) | New Simplified Arakawa-Schubert (SAS) [14] | Betts-Miller-Janjic [2] | Grell-Devenyi [93] |
| **Planetary Boundary Layer** | Bougeault-Lacarrere (BouLac) [8] | Bougeault-Lacarrere (BouLac) [8] | Yonsei University Scheme (YSU) [1] |
| **Longwave Radiation** | Geophysical Fluid Dynamics Laboratory (GFDL) [99] | Rapid Radiative Transfer Model for General Circulation Models (RRTMG) [4] | Rapid Radiative Transfer Model (RRTM) [1] |
| **Shortwave Radiation** | Geophysical Fluid Dynamics Laboratory (GFDL) [99] | Rapid Radiative Transfer Model for General Circulation Models (RRTMG) [4] | Dudhia [1] |
| **Land Surface** | Rapid Update Cycle (RUC) land-surface model [3] | Rapid Update Cycle (RUC) land-surface model [3] | Unified Noah land-surface model [2] |
| **Surface Layer** | Revised MM5 Monin-Obukhov [1] | Revised MM5 Monin-Obukhov [1] | Old MM5 surface layer [91] |
| **Simulation start date** | 09/11 00:00 UTC | 08/30 00:00 UTC | 08/25 06:00 UTC |





**Table A1.** Calibration performance summary. From Mure-Ravaud et al. (2017a) and Mure-Ravaud et al. (2017b) (unpublished manuscript).

|  |  | Hurricane Frances (2004) | Hurricane Ivan (2004) | Hurricane Isaac (2012) |
|---|---|---|---|---|
| Relative error |  | +12 % | +6.3 % | -13 % |
| $50^{th}$ percentile | % overlap | 85 % | 85 % | 56 % |
| threshold | PFAR | 1.12 | 1.01 | 0.87 |
| $75^{th}$ percentile | % overlap | 75 % | 65 % | 49 % |
| threshold | PFAR | 1.20 | 1.02 | 0.94 |
| $90^{th}$ percentile | % overlap | 62 % | 59 % | 53 % |
| threshold | PFAR | 1.33 | 1.36 | 0.84 |
| $95^{th}$ percentile | % overlap | 49 % | 59 % | 56 % |
| threshold | PFAR | 1.58 | 1.66 | 0.72 |
| $97.5^{th}$ percentile | % overlap | 44 % | 59 % | 40 % |
| threshold | PFAR | 1.82 | 2.04 | 0.79 |
| $99^{th}$ percentile | % overlap | 20 % | 62 % | 26 % |
| threshold | PFAR | 1.81 | 2.79 | 1.27 |