# Peer review of "Maximization of the precipitation from tropical cyclones over a target area through physically based storm transposition"

_Hydrology and Earth System Sciences, 2017_

## Short Comment (SC1) · 21 Feb 2018

I congratulate the authors for the innovative tropical storm transposition using the numerical weather model. I have been skeptical about the numerical model based PMP estimation for tropical storm regions. Although this methodology may be difficult to ensure maximizing precipitation in a target watershed, these numerical experiments are scientifically very interesting as well as potentially useful in practice. Therefore, I believe this manuscript should be accepted for publication in the HESS.

I have a few minor optional questions on this work: 1. Is there any risk of numerical instability due to the storm transposition? 2. What state variables did you modified?

It may be worth to list up the state variables in the WRF simulations in this article. 3. When the atmospheric state variables is interpolated, is there any note on the topography effect?

---

## Referee Comment (RC1) · Anonymous Referee #1 · 4 Mar 2018

This paper attempts to define a fully physically based method to estimate a maximum precipitation that would result from tropical cyclones over a given target area, which is more or less close to the effective landing area. This method is applied to four cyclone cases. As such, the subject of this paper is obviously of great interest for HESS.

The proposed "transposition method" relies on a series of steps that are rather precisely defined: (i) define the centre and radius of the cyclonic vertex, (ii) define the meteorological background field as being the field outside of the cyclonic vertex, as well as its linear interpolation inside of the vertex, (iii) define the perturbation field by subtracting the background field to the actual field, (iv) translate/shift the perturbation

field (v) linearly recombine it with the (fixed) background field, (v) run a regional atmospheric model (RAM) with the obtained initial conditions, (vi) estimate the resulting, accumulated precipitation over the target area and a given period of time (72 hours in the present study).

At first, references to the concept of Probable Maximum Precipitation (PMP) seem somewhat misleading: although the authors have been inspired by some techniques developed in PMP approaches, their goal is more precise as explicitly stated in the introduction and somewhat in the title of their paper. Furthermore, as discussed below, it seems that their study puts into question PMP rather than supports it.

Secondly, the claim that the present method is fully physically based is not obvious for at least two reasons: - whereas, the RAM can be considered as physically based on the subrange of the explicit scales, this is not the case for the parametrisation of the smallest scales that are essential for precipitation; - most other procedure steps are not physically based.

Furthermore, the linear nature of several steps (ii - iv, respectively subtracting, interpolating and adding the background field) are rather at odd with the nonlinearity of the system. It is also questionable to define the cyclonic vertex as a circle (step i), whereas the material contours of various fields are rather convoluted. A priori, these linear simplifications, as well as the parametrisation, may introduce non negligible model/method errors that should be acknowledged, despite they generate frustrations with respect to the applicative goal of the paper, namely the accuracy of the heavy precipitation estimates.

On the contrary, I believe that the authors should emphasise and promote a result of their study that is a consequence from the preserved nonlinearities of the systems. Indeed, they are right to observe and argue that these nonlinearities yield a complex sensitivity of the vertex track with respect to the initial translation of the vertex, in particular a small translation can be well sufficient to substantially modify the vertex track so

that it will go over the target area. Similar observations are done and could be further developed on uncertainties resulting from the choice of the simulation starting date, therefore of its initial conditions. In particular there is a sensitivity to the relative intensity of the perturbation field, which might interfere with the aforementioned method errors (i.e., highest intensities will presumably amplify these errors). The authors are right to mention a similar sensitivity to the choice of the RAM parametrisation settings. By the way, according to the rightest hand side equation of Eq.1, it seems the authors selected the hydrostatic option of WRF, whereas it is basically a non-hydrostatic RAM.

A priori, the above results and considerations have important implications on the accuracy of the estimates of the heavy precipitations over the target area, i.e., they presumably displays a much higher variability than expected. Does it require an ensemble approach and a statistical analysis of the extremes? Does the latter put into question PMP approaches? I believe that these questions should addressed, at least tentatively, by the authors. Overall, I believe that the paper should devote more room to the methodological questions and display a terser presentation of the study cases.

---

## Referee Comment (RC2) · Anonymous Referee #2 · 14 Mar 2018

General Comments:

The present study proposes a method to estimate the probable maximum precipitation over a specific river basin with the use of a regional meteorological model. The proposed method is physically based and transposes a TC location by separating the circulation associated with a TC and its background state. The way how the proposed method works is demonstrated for four hurricanes cases. In general, this type of approaches that relocate the initial position of a specific TC is useful for assessing the impacts of the TC hazards at basin scales, because, as the authors recognize, the precipitation pattern by a specific TC is critically dependent on the track of a TC as

well as the intensity of a TC. Thus, the present study potentially deals with a scientific important issue.

However, the scientific originality of the present study is doubtful. There are some studies that proposed a TC relocation approach by using a TC bogusing scheme. For example, Ishikawa et al. (2013) and Oku et al. (2014) proposed a TC bogusing scheme that uses potential vorticity to separate the TC field from the background flow. Their proposed approach has been applied for dynamical downscaling assessment of regional-scale precipitation induced by severe typhoons in the past and in the future climate simulations. The probable maximum precipitation has been estimated by searching a worst-case typhoon track. The recent reviews on this issue were provided in Mori and Takemi (2016) and Takemi et al. (2016). Considering these previous works, the originality of the present study is not well described. Please consult with these previous papers and the references therein. The sentence on page 3, line 29 "this is the first study investigating a fully physically based method ..." should be revised by incorporating some of the previous studies. Please articulate and emphasize the scientific merit in this study.

Another issue is the proposed method itself. The explanation described in Section 2 seems not to be clear. This reviewer does not understand how you would define the TC circulation from the background field. If you assume that a TC has an axisymmetric structure and that the TC circulation can be approximated as some type of analytical vortices (such as Rankine vortex), you could separate the TC circulation from the background field. But how would you determine TC-related relative humidity and temperature from the background? I think that there are some other assumptions; for example, the thermal wind balance should be assumed in order to derive TC-related temperature field. If you include moisture in the temperature definition in the form of virtual temperature, you could also derive TC-related moisture field. However, the current manuscript is lack of sufficient explanations on how to isolate TC field from the background. Furthermore, if there is a background flow, you may need to subtract

background flow field in order to obtain axisymmetric flow structure of the TC. In addition, the way to determine the radius R of the TC is not explained. Is this radius the radius of the maximum wind? If so, TC-related circulation outside the radius R should somehow be eliminated from the background.

Overall, although this reviewer understands the scientific importance dealt in the present study, the scientific originality is vague and the proposed method is not convincing. Major revisions must be conducted before considering the publication of the present study.

Technical Corrections:

1. The figures are not numbered in the order of their appearance. Please take special care when you revise.

2. Page 6, lines 8-10, "It spawned . . . and into the New England area (Fig.8c).": Fig. 8c does not include the New England area, which is misleading. Please reword.

3. Page 8, line 23, IVT: What does IVT mean? Please spell out.

4. Page. 10, line 32-34, " However, in the case of Hurricane Isaac . . . the maximized precipitation field is overall slightly less intense than the observed precipitation field.": Figs. 13 and 14 does not include a panel of the observed precipitation, and I cannot evaluate if this statement is correct or not. Please add the figure showing the observation field.

References:

Ishikawa, H., Y. Oku, S. Kim, T. Takemi, and J. Yoshino, 2013: Estimation of a possible maximum flood event in the Tone River basin, Japan caused by a tropical cyclone. Hydrological Processes, Vol. 27, pp. 3292-3300, doi: 10.1002/hyp.9830.

Oku, Y., J. Yoshino, T. Takemi, and H. Ishikawa, 2014: Assessment of heavy rainfall-induced disaster potential based on an ensemble simulation of Typhoon Talas (2011)

with controlled track and intensity. Natural Hazards and Earth System Sciences, Vol. 14, pp. 2699-2709, doi:10.5194/nhess-14-2699-2014.

Mori, N., and T. Takemi, 2016: Impact assessment of coastal hazards due to future changes of tropical cyclones in the North Pacific Ocean. Weather and Climate Extremes, Vol. 11, pp. 53-69, doi:10.1016/j.wace.2015.09.002.

Takemi, T., Y. Okada, R. Ito, H. Ishikawa, and E. Nakakita, 2016: Assessing the impacts of global warming on meteorological hazards and risks in Japan: Philosophy and achievements of the SOUSEI program. Hydrological Research Letters, Vol. 10, pp. 119-125, doi: 10.3178/hrl.10.119.

---

## Author Comment (AC1) · 10 Apr 2018

Comment:

N. Ohara

nohara1@uwyo.edu

I congratulate the authors for the innovative tropical storm transposition using the numerical weather model. I have been skeptical about the numerical model based PMP estimation for tropical storm regions. Although this methodology may be difficult to ensure maximizing precipitation in a target watershed, these numerical experiments are scientifically very interesting as well as potentially useful in practice. Therefore, I believe this manuscript should be accepted for publication in the HESS. I have a few minor optional questions on this work:

1. Is there any risk of numerical instability due to the storm transposition?

2. What state variables did you modified? It may be worth to list up the state variables in the WRF simulations in this article.

3. When the atmospheric state variables is interpolated, is there any note on the topography effect?

Response:

Thank you for your positive feedback. Regarding your questions:

1. Is there any risk of numerical instability due to the storm transposition?

The emergence of numerical instability due to the manipulation of the input data is a valid concern. In our case however, we did not observe any numerical instability due to the storm transposition. Indeed, we have endeavored to develop a transposition scheme which perturbs the initial conditions as little as possible. The storm is first separated from its background field and only this "perturbation" is transposed. Moreover, we insisted that the transposition exercise should ideally be performed when the tropical cyclone moves over the ocean, far from land, and before it starts its extratropical transition. Under these conditions the vortex is usually relatively small and seems to be advected by the large scale flow. Finally, although there is technically no limits regarding how far the tropical cyclone can be transposed from its initial location, we believe that this transposition exercise should remain a perturbation exercise: the nonlinearity in the system is taken advantage of so that a small perturbation of the initial location of the vortex brings sufficient changes to its track, enabling the storm to overlap the

target area. Numerical instabilities can certainly occur if the amount of shift is too large and if the tropical cyclone is implanted at a location where it has no legitimacy to be, which is the case for example if it is transposed from a location over the ocean to a location over land.

2. What state variables did you modified? It may be worth to list up the state variables in the WRF simulations in this article.

The state variables that were modified are:

1) Surface variables: skin temperature, temperature at 2 meters, relative humidity at 2 meters, wind speed at 10 meters, surface pressure, pressure at mean sea level;

2) Pressure level variables: temperature, wind speed, relative humidity, geopotential height.

Thank you for your suggestion: we will list up the state variables in the WRF simulations in the article.

3. When the atmospheric state variables is interpolated, is there any note on the topography effect?

State variables are interpolated over the ocean, so that topographic effects are usually negligible. However, some state variables such as the surface wind speed and the surface relative humidity are very sensitive to the transition from sea to land, which might raise some complications if the initial vortex is located near an island, as it was the case for Hurricane Isaac (see Figure 10c). In such a case, the background fields were interpolated only over the ocean, so that the perturbation fields did not carry with them unrealistic values. We note that this work is only a preliminary work to show that it is possible to estimate the probable maximum precipitation over a river basin subject to the effect of tropical cyclones in the eastern U.S. using a physically based approach.

---

## Author Comment (AC2) · 10 Apr 2018

**Comment:**

Anonymous Referee #1

This paper attempts to define a fully physically based method to estimate a maximum precipitation that would result from tropical cyclones over a given target area, which is more or less close to the effective landing area. This method is applied to four cyclone cases. As such, the subject of this paper is obviously of great interest for HESS.

The proposed "transposition method" relies on a series of steps that are rather precisely defined: (i) define the centre and radius of the cyclonic vertex, (ii) define the meteorological background field as being the field outside of the cyclonic vertex, as well as its linear interpolation inside of the vertex, (iii) define the perturbation field by subtracting the background field to the actual field, (iv) translate/shift the perturbation field (v) linearly recombine it with the (fixed) background field, (v) run a regional atmospheric model (RAM) with the obtained initial conditions, (vi) estimate the resulting, accumulated precipitation over the target area and a given period of time (72 hours in the present study).

At first, references to the concept of Probable Maximum Precipitation (PMP) seem somewhat misleading: although the authors have been inspired by some techniques developed in PMP approaches, their goal is more precise as explicitly stated in the introduction and somewhat in the title of their paper. Furthermore, as discussed below, it seems that their study puts into question PMP rather than supports it.

Secondly, the claim that the present method is fully physically based is not obvious for at least two reasons: - whereas, the RAM can be considered as physically based on the subrange of the explicit scales, this is not the case for the parametrisation of the smallest scales that are essential for precipitation; - most other procedure steps are not physically based.

Furthermore, the linear nature of several steps (ii - iv, respectively subtracting, interpolating and adding the background field) are rather at odd with the nonlinearity of the system. It is also questionable to define the cyclonic vertex as a circle (step i), whereas the material contours of various fields are rather convoluted. A priori, these linear simplifications, as well as the parametrisation, may introduce non

negligible model/method errors that should be acknowledged, despite they generate frustrations with respect to the applicative goal of the paper, namely the accuracy of the heavy precipitation estimates.

On the contrary, I believe that the authors should emphasise and promote a result of their study that is a consequence from the preserved nonlinearities of the systems. Indeed, they are right to observe and argue that these nonlinearities yield a complex sensitivity of the vertex track with respect to the initial translation of the vertex, in particular a small translation can be well sufficient to substantially modify the vertex track so that it will go over the target area. Similar observations are done and could be further developed on uncertainties resulting from the choice of the simulation starting date, therefore of its initial conditions. In particular there is a sensitivity to the relative intensity of the perturbation field, which might interfere with the aforementioned method errors (i.e., highest intensities will presumably amplify these errors). The authors are right to mention a similar sensitivity to the choice of the RAM parametrisation settings. By the way, according to the rightest hand side equation of Eq.1, it seems the authors selected the hydrostatic option of WRF, whereas it is basically a non-hydrostatic RAM.

A priori, the above results and considerations have important implications on the accuracy of the estimates of the heavy precipitations over the target area, i.e., they presumably displays a much higher variability than expected. Does it require an ensemble approach and a statistical analysis of the extremes? Does the latter put into question PMP approaches? I believe that these questions should addressed, at least tentatively, by the authors. Overall, I believe that the paper should devote more room to the methodological questions and display a terser presentation of the study cases.

**Response:**

Comment: At first, references to the concept of Probable Maximum Precipitation (PMP) seem somewhat misleading: although the authors have been inspired by some techniques developed in PMP approaches, their goal is more precise as explicitly stated in the introduction and somewhat in the title of their paper. Furthermore, as discussed below, it seems that their study puts into question PMP rather than supports it.

Response: The objective of this study is to prove the feasibility of using a physically based approach for the estimation of the PMP over a target basin in the eastern U.S. subject to the effects of tropical

cyclones (TCs). It is true that the outcomes from this study raise serious questions regarding the legitimacy and validity of traditional PMP approaches in the case of TCs.

Comment: Secondly, the claim that the present method is fully physically based is not obvious for at least two reasons: - whereas, the RAM can be considered as physically based on the subrange of the explicit scales, this is not the case for the parametrisation of the smallest scales that are essential for precipitation; - most other procedure steps are not physically based.

Response: The authors recognize that the method presented in this paper is not "fully" physically based for the reasons provided by the reviewer. As such, the word "fully" will be removed from the revised version of the manuscript. The reason why we did not use a fully physically based approach is that such an approach does not exist yet. The Weather Research and Forecasting (WRF) model is one of the most advanced and physically based tools available to date, and this is why we used this model in our study. Yet, to the authors' knowledge, the storm transposition method proposed in this article remains much more physically based than any other PMP estimation method proposed so far for the eastern U.S. As models and in particular the representation of the smallest scales keep improving, the legitimacy of using a regional atmospheric model (RAM) for the purpose of PMP estimation will become stronger and the benefits of this approach more obvious.

As far as the procedure steps are concerned, it is true that the vortex relocation procedure presented in this article which simply shifts the perturbation fields obtained by subtracting the background fields from the initial fields is less physically based than other preexisting vortex relocation procedures such as the vortex bogusing techniques used in forecasting.

Finally, the authors note that no cumulus parameterization was used in the simulation inner domain (see Table 1 in the article).

Comment: Furthermore, the linear nature of several steps (ii - iv, respectively subtracting, interpolating and adding the background field) are rather at odd with the nonlinearity of the system.

Response: The vortex relocation procedure presented in this study is relatively simple. The goal of this preliminary study is to prove the feasibility of a physically based approach for the storm transposition of

TCs in the Atlantic basin. More sophisticated tools are available and will be investigated in the future. For example, vortex "bogusing" is a technique that is widely employed by the forecasting community. They observed that the discrepancies between the coarse resolution of the analysis and the fine resolution of the hurricane model can cause a significant period of vortex adjustment at the beginning of the forecast which may have very prejudicial effects on the quality of the forecasts, both in terms of the storm's track and intensity. As a result, several studies (e.g. Kurihara et al., 1993; Zou and Xiao, 2000; Hsiao et al., 2010) have proposed vortex removal procedures, vortex relocation procedures, etc. which are in general more sophisticated than the method proposed in our study. The objective of these procedures is to provide a TC in the initial condition which is as realistic as possible and which is physically and thermodynamically consistent with the background fields and with the model's physics. Recent studies (e.g. Zou and Xiao, 2000) have used 4 dimensional variational bogus data assimilation (bogus 4D-Var) to provide a physically and thermodynamically consistent initial vortex. One difference between the aforementioned studies and our approach is that they first implant a simple axisymmetric bogus vortex (based for example on the gradient wind equations) before recovering the vortex asymmetry through different means such as 4D-Var, whereas we transposed and implanted the full vortex obtained by subtracting the interpolated background fields from the original fields. While the aforementioned vortex adjustment at the beginning of the simulation is detrimental for the purpose of forecasting, the authors believe that it is less of a problem for the purpose of design and in particular PMP estimation. Indeed, during this spinup time the RAM corrects the inconsistencies that may be present in the initial condition due to the pointed linear nature of the vortex relocation steps since the RAM numerically solves the nonlinear equations governing the conservations of mass, momentum and energy.

Furthermore, the authors emphasized in the article that several precautions need to be taken while performing the transposition exercise. Section 2 stresses that the transposition exercise should be ideally performed when the TC is moving over the ocean, far from land, and before its extratropical transition. In this case, the TC is usually relatively small and seems to be simply advected by the large scale flow.

Comment: It is also questionable to define the cyclonic vertex as a circle (step i), whereas the material contours of various fields are rather convoluted.

Response: The interpolation region was defined as a circle, not the cyclonic vortex per se. This is because of the general tendency of TCs to be somehow circular in shape. We acknowledge that the material

contours of the various fields may be complicated and the fields asymmetric with respect to the storm center. Such large asymmetry is particularly likely to be encountered when the TC moves within the midlatitudes and undergoes its extratropical transition (Chan and Kepert, 2010). On the other hand, we stressed that the transposition exercise should be performed before the TC starts its extratropical transition, and ideally when it is moving over the ocean and far from land. In this case, it is likely that the material contours of the various fields should be close to circular in shape. Actually, whatever shape may be adopted for the interpolation region. We chose a circle because this is the shape that seemed the most reasonable to us given the nature of the storm under investigation.

Comment: A priori, these linear simplifications, as well as the parametrisation, may introduce non negligible model/method errors that should be acknowledged, despite they generate frustrations with respect to the applicative goal of the paper, namely the accuracy of the heavy precipitation estimates.

Response: Do the modifications brought to the initial conditions generate "errors" properly speaking in the output of the model? We do not think so. First of all, as mentioned earlier in this discussion, the RAM smoothes out the inconsistencies that may exist in the initial fields due to the nature of our vortex relocation procedure since it numerically solves the nonlinear equations governing the conservation of the mass, momentum and energy. We recall that the simulation start date is taken several days before the TC makes landfall, so that by the time of landfall and especially by the time the most intense rainfall is produced over the target area, all the fields produced by the RAM are fully consistent physically and thermodynamically.

Moreover, the authors believe that thinking in these terms is giving to much importance to the initial condition. Indeed, does the structure and intensity of the precipitation field depend more on the detailed structure of the initial vortex, or does it depend more on how the TC interacts with its environment between the simulation start date and the time of intense precipitation? Figure R1 below shows the evolution of the total precipitable water (PW) in Hurricane Frances with a 24-h time increment for the simulations corresponding to Fig. 12b and 12c in the article. Note that in this figure Hurricane Frances is located east of Puerto Rico (see also Fig. 10b in the article). We recall that only about 8 km separate the location of the initial vortices between one simulation and the other whereas the locations of landfall defer by more than 100 km. The right column in Fig. R1 shows the difference between the two. We observe that the trajectories are almost identical until Hurricane Frances approaches Florida,

after which they start to diverge rapidly, with the hurricane affecting mainly Florida, Georgia and the western Carolinas in one case versus Alabama and Mississippi in the other case. It is clear in this example that the differences in the locations, structures, and intensities of the precipitation fields between one simulation and the other are due to the effects accumulated during the storm motion rather than to the details of how one initial vortex's structure defers from the other initial vortex's structure. Although the initial vortices are very close to each other, the trajectories diverge, first very slowly as the storm moves over the ocean, then much faster as the storm approaches land. In a nutshell, we think that differences between one precipitation field and another are more a function of how the TC interacts with its environment including how the moisture is advected over the region, how it interacts with the local topography and how its converges and diverges (see Fig. 9 in the article) rather than being a function of the details of how one initial vortex's structure could have been or should have been if one had used a more sophisticated vortex relocation procedure than the one used in this study.

Second, let us discuss the parameterization. Instead of "errors", the authors prefer to discuss uncertainties. This relates to a point that the reviewer raises later when proposing an ensemble approach. The traditional PMP approaches are essentially deterministic and do not in general offer any way to estimate the uncertainties associated with the PMP estimate. Although it is true that the uncertainties associated with the model's options (parameterization), with the initial condition (in particular the simulation start date), and with the dataset used to provide the initial and boundary conditions cause the PMP estimate to be also uncertain, such uncertainty can be quantified by using an ensemble approach as discussed in Section 5 of the article. The physically based PMP estimation exercise should be performed with as many combinations of the model's options (as long as each combination has been beforehand validated as discussed in the appendix), simulations start dates, and datasets for initial/boundary conditions.

Finally, although it is true that some errors (or uncertainties depending on the point of view) are associated with the physically based PMP estimate, one should not forget that a tremendous advantage of this approach is that the RAM provides the underlying mechanisms responsible for generating the extreme precipitation such as moisture transport and convergence, topographical effects, etc. In other words, although it is true that the RAM may slightly overestimate or underestimate the maximum precipitation over the target basin because of the choice of the vortex relocation procedure and the choice of the parameterization schemes, the additional information that it provides on the underlying fields responsible for causing the extreme rainfall such as integrated vapor transport, precipitable water,

temperature and wind field as well as the temporal and spatial structures of these fields remains extremely valuable.

Comment: On the contrary, I believe that the authors should emphasise and promote a result of their study that is a consequence from the preserved nonlinearities of the systems. Indeed, they are right to observe and argue that these nonlinearities yield a complex sensitivity of the vertex track with respect to the initial translation of the vertex, in particular a small translation can be well sufficient to substantially modify the vertex track so that it will go over the target area. Similar observations are done and could be further developed on uncertainties resulting from the choice of the simulation starting date, therefore of its initial conditions.

Response: The goal of this study is to show the feasibility of a physically based TC transposition approach for the purpose of PMP estimation in the eastern U.S. As pointed by the reviewer, investigating the uncertainties resulting from the choice of the simulation starting date, therefore of its initial conditions is an essential issue and will be tackled in a future study.

Comment: In particular there is a sensitivity to the relative intensity of the perturbation field, which might interfere with the aforementioned method errors (i.e., highest intensities will presumably amplify these errors). The authors are right to mention a similar sensitivity to the choice of the RAM parametrisation settings.

Response: The precipitation field generated by the TC indeed depends on the relative intensity of the perturbation field. In particular, a small change in the structure of the initial vortex (not only its position) may result in different trajectories because the consequences of this change can amplify over time in a nonlinear way. However, is it appropriate to say that this is going to cause an "error" in the precipitation field generated by the TC? We do not think so. Indeed the atmospheric fields produced by the RAM are fully physically and thermodynamically consistent since the RAM solves the equations governing the conservation of mass, momentum, and energy. To this extent, how can they be wrong (as long as the

model is provided with realistic initial conditions)? It might be more appropriate to interpret the precipitation fields produced by the RAM as different realizations of what a given TC could have caused. This is actually the idea behind the method proposed in the article: generating realizations of a given TC by perturbing the location of the storm in the initial conditions, with a focus on those realizations producing extreme precipitation over the target area.

Comment: By the way, according to the rightest hand side equation of Eq.1, it seems the authors selected the hydrostatic option of WRF, whereas it is basically a non-hydrostatic RAM.

Response: The hydrostatic assumption was used to estimate the integrated vapor transport (IVT) from the other fields produced by the WRF model since the WRF model does not output the IVT. We did not use the hydrostatic option of WRF for the simulations.

Comment: A priori, the above results and considerations have important implications on the accuracy of the estimates of the heavy precipitations over the target area, i.e., they presumably displays a much higher variability than expected. Does it require an ensemble approach and a statistical analysis of the extremes?

Response: As discussed previously, Section 5 in the article clearly encourages the adoption of an ensemble approach for the physically based PMP estimation in order to estimate the uncertainties associated with the model's options (parameterization), simulation start date, and dataset used for initial/boundary conditions.

Comment: Does the latter put into question PMP approaches?

Response: The latter indeed puts into question the traditional PMP approaches.

Comment: I believe that these questions should addressed, at least tentatively, by the authors. Overall, I believe that the paper should devote more room to the methodological questions and display a terser presentation of the study cases.

Response: The objective of this study is to show the feasibility of a physically based TC transposition approach for the purpose of PMP estimation in the eastern U.S. A more detailed investigation of the uncertainties associated with the parameterization, simulation start date, and dataset for initial/boundary conditions will be performed in a later study. As far as the room to the methodological questions is concerned, the authors believe that this article and in particular Section 2 already provides sufficient details on the procedure used to shift the TC in the initial condition.

[Figure]

Figure R1 - Left column: evolution of the total precipitable water (mm) in Hurricane Frances with a 24-h time increment starting on 08/30/2004 00:00 UTC for the simulation corresponding to Fig. 12b in the article. Middle column: same as left column but for the simulation corresponding to Fig. 12c in the article. Right column: difference between middle column and left column.

[Figure]

Figure R1 – continued

[Figure]

Figure R1 - continued

**References**

Chan, J.C. and Kepert, J.D., 2010. Global perspectives on Tropical cyclones: From science to mitigation, 4. World Scientific.

Hsiao, L.-F. et al., 2010. A vortex relocation scheme for tropical cyclone initialization in advanced research WRF. Monthly Weather Review, 138(8): 3298-3315.

Kurihara, Y., Bender, M.A. and Ross, R.J., 1993. An initialization scheme of hurricane models by vortex specification. Monthly weather review, 121(7): 2030-2045.

Zou, X. and Xiao, Q., 2000. Studies on the initialization and simulation of a mature hurricane using a variational bogus data assimilation scheme. Journal of the atmospheric sciences, 57(6): 836-860.

---

## Author Comment (AC3) · 10 Apr 2018

**General Comments:**

The present study proposes a method to estimate the probable maximum precipitation over a specific river basin with the use of a regional meteorological model. The proposed method is physically based and transposes a TC location by separating the circulation associated with a TC and its background state. The way how the proposed method works is demonstrated for four hurricanes cases. In general, this type of approaches that relocate the initial position of a specific TC is useful for assessing the impacts of the TC hazards at basin scales, because, as the authors recognize, the precipitation pattern by a specific TC is critically dependent on the track of a TC as well as the intensity of a TC. Thus, the present study potentially deals with a scientific important issue.

However, the scientific originality of the present study is doubtful. There are some studies that proposed a TC relocation approach by using a TC bogusing scheme. For example, Ishikawa et al. (2013) and Oku et al. (2014) proposed a TC bogusing scheme that uses potential vorticity to separate the TC field from the background flow. Their proposed approach has been applied for dynamical downscaling assessment of regional-scale precipitation induced by severe typhoons in the past and in the future climate simulations. The probable maximum precipitation has been estimated by searching a worst-case typhoon track. The recent reviews on this issue were provided in Mori and Takemi (2016) and Takemi et al. (2016). Considering these previous works, the originality of the present study is not well described. Please consult with these previous papers and the references therein. The sentence on page 3, line 29 "this is the first study investigating a fully physically based method ..." should be revised by incorporating some of the previous studies. Please articulate and emphasize the scientific merit in this study. Another issue is the proposed method itself. The explanation described in Section 2 seems not to be clear. This reviewer does not understand how you would define the TC circulation from the background

field. If you assume that a TC has an axisymmetric structure and that the TC circulation can be approximated as some type of analytical vortices (such as Rankine vortex), you could separate the TC circulation from the background field. But how would you determine TC-related relative humidity and temperature from the background? I think that there are some other assumptions; for example, the thermal wind balance should be assumed in order to derive TC-related temperature field. If you include moisture in the temperature definition in the form of virtual temperature, you could also derive TC-related moisture field. However, the current manuscript is lack of sufficient explanations on how to isolate TC field from the background. Furthermore, if there is a background flow, you may need to subtract background flow field in order to obtain axisymmetric flow structure of the TC. In addition, the way to determine the radius R of the TC is not explained. Is this radius the radius of the maximum wind? If so, TC-related circulation outside the radius R should somehow be eliminated from the background.

Overall, although this reviewer understands the scientific importance dealt in the present study, the scientific originality is vague and the proposed method is not convincing. Major revisions must be conducted before considering the publication of the present study.

Technical Corrections:

1. The figures are not numbered in the order of their appearance. Please take special care when you revise.

2. Page 6, lines 8-10, "It spawned … and into the New England area (Fig.8c).": Fig. 8c does not include the New England area, which is misleading. Please reword.

3. Page 8, line 23, IVT: What does IVT mean? Please spell out.

4. Page. 10, line 32-34, " However, in the case of Hurricane Isaac … the maximized precipitation field is overall slightly less intense than the observed precipitation field.": Figs. 13 and 14 does

not include a panel of the observed precipitation, and I cannot evaluate if this statement is correct or not. Please add the figure showing the observation field.

References:

Ishikawa, H., Y. Oku, S. Kim, T. Takemi, and J. Yoshino, 2013: Estimation of a possible maximum flood event in the Tone River basin, Japan caused by a tropical cyclone. Hydrological Processes, Vol. 27, pp. 3292-3300, doi: 10.1002/hyp.9830.

Oku, Y., J. Yoshino, T. Takemi, and H. Ishikawa, 2014: Assessment of heavy rainfallinduced disaster potential based on an ensemble simulation of Typhoon Talas (2011) with controlled track and intensity. Natural Hazards and Earth System Sciences, Vol. 14, pp. 2699-2709, doi:10.5194/nhess-14-2699-2014.

Mori, N., and T. Takemi, 2016: Impact assessment of coastal hazards due to future changes of tropical cyclones in the North Pacific Ocean. Weather and Climate Extremes, Vol. 11, pp. 53-69, doi:10.1016/j.wace.2015.09.002.

Takemi, T., Y. Okada, R. Ito, H. Ishikawa, and E. Nakakita, 2016: Assessing the impacts of global warming on meteorological hazards and risks in Japan: Philosophy and achievements of the SOUSEI program. Hydrological Research Letters, Vol. 10, pp. 119-125, doi: 10.3178/hrl.10.119.

**Responses:**

Comment:

However, the scientific originality of the present study is doubtful. There are some studies that proposed a TC relocation approach by using a TC bogusing scheme. For example, Ishikawa et al. (2013) and Oku et al. (2014) proposed a TC bogusing scheme that uses potential vorticity to separate the TC field from the background flow. Their proposed approach has been applied for dynamical downscaling assessment of regional-scale precipitation induced by severe typhoons in the past and in the future climate simulations. The probable maximum precipitation has been estimated by searching a worst-case typhoon track. The recent reviews on this issue were provided in Mori and Takemi (2016) and Takemi et al. (2016). Considering these previous works, the originality of the present study is not well described. Please consult with these previous papers and the references therein. The sentence on page 3, line 29 "this is the first study investigating a fully physically based method …" should be revised by incorporating some of the previous studies. Please articulate and emphasize the scientific merit in this study.

Response:

The authors were not aware of these studies and thank the reviewer for bringing them to their knowledge. These studies will be acknowledged in the revised manuscript. The sentence "this is the first study investigating a fully physically based method …" as well as mentions to "a new physically based storm transposition" will be revised.

Comment:

Another issue is the proposed method itself. The explanation described in Section 2 seems not to be clear. This reviewer does not understand how you would define the TC circulation from the background field. If you assume that a TC has an axisymmetric structure and that the TC circulation can be approximated as some type of analytical vortices (such as Rankine vortex), you could separate the TC circulation from the background field. But how would you determine TC-related relative humidity and temperature from the background? I think that there are some

other assumptions; for example, the thermal wind balance should be assumed in order to derive TC-related temperature field. If you include moisture in the temperature definition in the form of virtual temperature, you could also derive TC-related moisture field. However, the current manuscript is lack of sufficient explanations on how to isolate TC field from the background. Furthermore, if there is a background flow, you may need to subtract background flow field in order to obtain axisymmetric flow structure of the TC. In addition, the way to determine the radius R of the TC is not explained. Is this radius the radius of the maximum wind? If so, TC-related circulation outside the radius R should somehow be eliminated from the background.

Response:

The method used to separate the TC circulation from the background field in this study is very simple. The radius R of a TC is established by visual examination of the different fields and in particular of the wind field. In general, looking at the wind field (see the example given in Fig. 4), it is relatively easy to see where is the TC and what is its size. In this study, we did not use any automatic procedure to establish the TC's radius but did it manually for each TC. Once the radius R was established, the inside of the circle of radius R was cut off, and the fields inside the circle recovered by interpolation of the background fields. The perturbation fields obtained by subtracting the interpolated fields from the original fields were then shifted and added back to the background fields. The state variables that we modified are:

1) Surface variables: skin temperature, temperature at 2 meters, relative humidity at 2 meters, wind speed at 10 meters, surface pressure, pressure at mean sea level

2) Pressure level variables: temperature, wind speed, relative humidity, geopotential height.

The authors recognize that the vortex bogusing scheme used in the studies referred to by the reviewer is more involved than what was done in the current study. In particular, recovering the different fields by inversion of the potential vorticity insures to obtain a vortex in the initial

conditions which is fully consistent from a physical and thermodynamical point of view. Vortex bogusing has been widely used for forecasting purposes for this reason: it is essential in the case of forecasting to have the best possible estimation of the initial vortex so as to not miss the storm's trajectory. Is the case of engineering design such as for the estimation of PMP, as important as in the case of forecasting? The authors do not think so. The linear assumption made by adding the shifted perturbation fields to the background fields while the governing equations are nonlinear is an approximation which affects only the initial conditions. The nonlinearity of the system is then fully accounted for by integration of the Regional Atmospheric Model (RAM).

Answering the reviewer's questions:

1) the TC-related relative humidity and temperature were not obtained from the other fields: the method described previously to obtain the perturbation fields was also applied to the relative humidity and to the temperature.

2) R is not the radius of maximum wind. As explained previously, R was established by visual examination of the different fields.

The reviewer is asking for major revisions, without clearly explaining what points need to be revised in a major way.

1) As far as the originality of the study is concerned, the revised version of the manuscript will clearly acknowledge the merits of the existing studies for the shifting of typhoons over Japan. Yet, to the authors' knowledge, this study remains the first to use a physically based TC transposition procedure for Tropical Cyclones in the Atlantic basin.

2) It took months to perform the simulations presented in this manuscript. Although the approximation made in how the perturbation field is estimated and then added to the

background field result in an initial vortex that is not fully consistent from a physical and thermodynamical point of view, these inconsistencies are subsequently smoothed out by the RAM since it numerically solves the nonlinear equations governing the conservation of the mass, momentum and energy. Fig. R1 below shows the evolution of the total precipitable water (PW) in Hurricane Ivan during the 6 first hours of the simulation with an hourly time increment. The PW is constructed from the WRF outputs, and more precisely using the pressure field, specific humidity field, potential temperature field, and geopotential field. Besides, since the PW indicates how much moisture is contained in an atmospheric column, it plays a central role in the study of intense precipitation. If there was any particular aberration in the aforementioned fields due to our vortex relocation procedure, this would have detrimental consequences on the PW and would be observable in Fig. R1. The middle column corresponds to no shift whereas the left column corresponds to an amount of shift of 0.8° W and 3.6° S and the right column to an amount of shift of 0.8° E and 3.6° N. As can be observed in Fig. R1 no aberration appears in the PW: the TC which corresponds to the circular region containing large values of PW off the coasts of French Guiana and Suriname is shifted as expected.

The objective of this work is to show that it is possible to use a regional atmospheric model to offer a physically based approach to the estimation of PMP in the eastern U.S. This paper presents a preliminary study showing the feasibility of such an approach. The authors understand the reviewer's concerns regarding the fact that the initial fields are not completely consistent physically and thermodynamically. There is no claim in this study that the vortex relocation procedure presented herein is the right one and the one that should be used in future physically based PMP investigations. This vortex relocation procedure is very simple. It gives an approximation of the initial fields after transposition of the TC which obviously is not as legitimate as in the case of the studies referred to by the reviewer, for which the potential vorticity inversion method guarantees to obtain more consistent fields in the initial condition. Still, as Fig. R1 shows, the vortex relocation procedure presented herein does the job: the TC is shifted as expected. The simulation start dates are taken early enough so that the model smoothes out the inconsistencies that may exist in the initial conditions. By the time of landfall,

the different fields including the precipitation field are fully consistent physically and thermodynamically since the RAM numerically solves the nonlinear equations governing the conservation of mass, momentum and energy.

**Comment:**

1. The figures are not numbered in the order of their appearance. Please take special care when you revise.

**Response:**

It is not clear to the authors why the reviewer comments that the figures are not numbered in the order of their appearance. Indeed the manuscript was written and compiled with Latex so that the figures have to be numbered in the order of their appearance in the document, which is the case.

Now it is true that the figures do not always appear in the order of their appearance in the text. For example the first figure that is referred to as Fig. 4 is for the illustration of the transposition procedure. This is because this figure is discussed in more detail in section 3 on the transposition of Hurricane Ivan than it is in Section 2, and as such, according to the authors, it has more legitimacy to appear later in the document. However, in order to address the reviewer's comment, Fig. 4 will be placed at the beginning of the article and will become Fig. 1.

In the same way, Fig. 8c for the observed precipitation field in Hurricane Ivan is referred to at the beginning of Section 2 but appears later in the manuscript because it is further discussed when analyzing the maximized precipitation field in Hurricane Ivan. Since Fig. 8c belongs to a panel plot, it will not be possible to move this figure from Section 3 to Section 2.

Apart from these two cases, the authors do not see any aberrations in the way the figures are ordered.

Comment:

2. Page 6, lines 8-10, "It spawned … and into the New England area (Fig.8c).": Fig. 8c does not include the New England area, which is misleading. Please reword.

Response:

Figure 8c does not include the New England area because Fig. 8c shows the observed precipitation field inside the simulation inner domain and the simulation inner domain does not contain the New England area since it has been chosen based on the location of the target watershed located in western North Carolina. The sentence referred to by the reviewer aims at giving some historical information and context to Hurricane Ivan. In order to address the reviewer's comment, "(Fig. 8c)" will be moved from the end of the sentence to before "and into the New England area", so that the sentence becomes: "It spawned heavy precipitation ranging from 3-7 in depth along a large swath from Alabama and the Florida panhandle northeastward across the eastern Tennessee Valley (Fig. 8c) and into the New England area".

Comment:

3. Page 8, line 23, IVT: What does IVT mean? Please spell out.

Response:

The meaning of IVT is given on Page 8 line 18.

4. Page. 10, line 32-34, " However, in the case of Hurricane Isaac … the maximized precipitation field is overall slightly less intense than the observed precipitation field.": Figs. 13 and 14 does not include a panel of the observed precipitation, and I cannot evaluate if this statement is correct or not. Please add the figure showing the observation field.

The observed precipitation field for Hurricane Isaac is given in Figure 16c.

[Figure]

Figure R1: Evolution of the total precipitable water in Hurricane Ivan during the 36 first hours of the simulation with a 6-hourly time increment. The left column corresponds to an amount of shift of 0.8° W and 3.6° S, the middle column to no shift, and the right column to an amount of shift of 0.8° E and 3.6° N. Note that the TC over Florida is not Hurricane Ivan. Hurricane Ivan is located off the coasts of French Guiana and Suriname.